# Alternatively spliced STIM2.3 is an evolutionarily late store-operated Ca$^{2+}$ entry regulator expressed in brain

Vanessa Poth[1], Hoang Thu Trang Do[2], Lukas Jarzembowski[1], Katrin-Lisa Laius[1], Kathrin Förderer[1], Thomas Tschernig[3], Hanah B. Robertson[2], Dalia Alansary[1], Reza Shaebani[4,5], Volkhard Helms[2] and Barbara A. Niemeyer[1,5,*]

## ABSTRACT

Ca$^{2+}$ homeostasis is essential for cellular functions, with regulation by store-operated Ca$^{2+}$ entry (SOCE) omnipresent. Due to a lower affinity for endoplasmic reticulum (ER)-luminal Ca$^{2+}$, STIM2 regulates basal cytosolic Ca$^{2+}$ but also increases interaction and activation of ORAI proteins at ER–plasma membrane junctions after stimulation, whereas STIM1 requires stronger store depletion. In brain, STIM2 is highly expressed in hippocampal neurons. Here, we describe a short STIM2 splice variant, STIM2.3 (also known as STIM2G), that is present only in Old World monkeys, apes and humans, with expression mostly in brain. In contrast to other variants and despite lack of the polybasic domain, expression of STIM2.3 increased SOCE. Structure–function analysis delineated the role of the C-terminal motifs of STIM2 for Ca$^{2+}$ entry as well as for basal and induced activation of the NFAT transcription factor NFATc1. STIM2.3 displayed reduced interaction with AMPK and with activated AMPK. Neuronal expression of STIM2.3, in comparison to STIM2.2, increased the size of dendritic spine heads, suggesting a specific regulatory role in spine maintenance. Regulated splicing of STIM2.3 in brain might present a rapid mechanism to increase STIM2-mediated effects on gene expression, spine morphology or spontaneous excitability, potentially facilitating an evolutionarily recent expansion of brain complexity.

KEY WORDS: SOCE, Splicing, NFAT, AMPK, Dendritic spines, Neuron

## INTRODUCTION

Alternative splicing takes place in ∼95% of human genes (Pan et al., 2008; Wang et al., 2008). With the rapid development of RNA microarrays, bulk RNA sequencing (RNA seq) and single-cell RNA seq technologies, the complexity of regulated and dynamic alternative splicing is just beginning to be understood (Mazille et al., 2022; Siller et al., 2022; Ule and Blencowe, 2019), and it is becoming increasingly evident that resulting alternative protein functions can shape both the physiology and pathophysiology of organisms (Marasco and Kornblihtt, 2022). Adding or deleting protein domains due to alternative splicing can alter infrared sensing in bats (Gracheva et al., 2011) and sex determination in fruit flies (Bell et al., 1991), as well as change cell proliferation, and induce cancer and neurological disorders (Marasco and Kornblihtt, 2022). So perhaps it is not surprising that a ubiquitous mechanism essential for regulation of cellular Ca$^{2+}$ homeostasis and transcription factor activation is subject to cell type-specific alternative splicing. Store-operated Ca$^{2+}$ entry (SOCE) is triggered when activation of cell surface receptors induces depletion of Ca$^{2+}$ from the endoplasmic reticulum (ER). The decrease in luminal Ca$^{2+}$ concentration is differentially sensed by STIM proteins, with STIM2 – having a lower EF hand Ca$^{2+}$ affinity – responding to smaller changes in intraluminal Ca$^{2+}$, whereas STIM1 requires more substantial depletion (Brandman et al., 2007; Stathopulos et al., 2009). Activated STIM1 molecules unfold and prominently track along microtubules via association with end-binding proteins (EBs) to gather at ER–plasma membrane (PM) junctions, with the C-terminal polybasic domains (PBDs) as well as internal domains of the channel-activating domain (CAD/SOAR domain) attaching to inner leaflet phospholipids of the PM and with the CAD/SOAR domain clustering and activating ORAI ion channels residing in the PM (Chang et al., 2018; Cohen et al., 2023; Gulyas et al., 2022; Jermy, 2008; Smyth et al., 2007; reviewed, for example, in Prakriya and Lewis, 2015). The resulting elevated changes in intracellular Ca$^{2+}$ concentrations trigger, among other enzymes, activation of transcription factors such as nuclear factor of activated T cells (NFAT) proteins and NF-κB, with distinct differences in the amount of activation depending on the spatiotemporal profiles of Ca$^{2+}$ entry (Berry et al., 2018; Feske, 2007; Kar et al., 2016, 2011). While the role of NFATc1 and NF-κB activation is best understood in immune cells, and failure of NFATc1 activation results in severe immune deficiencies (Feske, 2019; Vaeth et al., 2020), its relevance is also seen in the brain, where SOCE-mediated NFAT1–NOX2 (CYBB)–NLRP1 inflammasome activation contributes to neuronal damage (Sun et al., 2022), and SOCE inhibition improves the outcome of traumatic brain injury and microglia-induced neuronal death (Mizuma et al., 2019; see also recent reviews Korshunov and Prakriya, 2025; Novakovic and Prakriya, 2025).

STIM genes (*STIM1* and *STIM2*) have a related genomic structure with 12 conventional exons (Williams et al., 2001) and long introns, spanning ∼220 kb of genomic DNA harboring a number of alternative exons (reviewed in Korshunov and Prakriya, 2025; Niemeyer, 2016). Thus, by utilizing alternative splicing, SOCE can

[1]Molecular Biophysics, Center for Integrative Physiology and Molecular Medicine (CIPMM), Bld. 48, Saarland University, Campus Homburg, Homburg 66421, Germany. [2]Center for Bioinformatics, Saarland University, Campus Saarbruecken, Saarbruecken 66123, Germany. [3]Institute of Anatomy and Cell Biology, Saarland University, Campus Homburg, Homburg 66421, Germany. [4]Department of Theoretical Physics, Saarland University, Campus Saarbruecken, Saarbruecken 66123, Germany. [5]Center for Biophysics, Saarland University, Campus Saarbruecken, Saarbruecken 66123, Germany.

*Author for correspondence (barbara.niemeyer@uks.eu)

V.P., 0000-0001-8813-9290; L.J., 0000-0002-7231-6106; T.T., 0000-0002-7788-1796; D.A., 0000-0002-7541-6057; R.S., 0000-0001-8587-6949; V.H., 0000-0002-2180-9154; B.A.N., 0000-0002-6963-0575

be adapted to suit cell type-specific needs. The first described *STIM1* splice variant shows a spliced extension of the conventional exon 11, leading to the longer protein variant STIM1L found in skeletal muscle (Darbellay et al., 2011). A dramatic switch in canonical STIM2 (from here on referred to as STIM2.2) function can be seen with alternative exon inclusion of 24 nucleotides into the region encoding the ORAI ion channel-activating domain (the CAD/SOAR domain), where splice inclusion reverts STIM2 from being a channel activator to an inhibitor of channel function (Miederer et al., 2015; Rana et al., 2015). Indeed, the alternative variant (STIM2.1, also known as STIM2β) is highly expressed in muscle, where expression of the splice variant regulates myogenesis by controlling SOCE-dependent transcription factors (Kim et al., 2019). Besides these two variants, we have recently described two additional alternative variants of STIM1, namely STIM1A and STIM1B, which, despite modifying SOCE to a smaller degree compared to STIM2.1, can profoundly influence the efficacy of synaptic transmission in a frequency-dependent manner in the case of the neuronal-specific STIM1B variant (Ramesh et al., 2021), or differentially affect gating of ORAI1, the STIM1 interactome, the local lipid environment and NFATc1 translocation (in the case of the more broadly expressed STIM1A; Knapp et al., 2022). This same variant (STIM1A) has also been described as STIM1β and has been shown to alter glioblastoma cell proliferation and affect wound closure, suggesting a differential function in cancer growth (Xie et al., 2022), although both reports (Knapp et al., 2022; Xie et al., 2022) postulate different molecular mechanisms leading to altered cellular function.

Following up on the described neuron-specific STIM1 splice variant (Ramesh et al., 2021) and the finding that STIM2.2 is prominently expressed in selected brain regions (i.e. hippocampus) – where it contributes to hypoxia induced neuronal death (Berna-Erro et al., 2009) but also stabilizes dendritic spine formation, protects spines from amyloid synaptotoxicity (Popugaeva et al., 2015; Sun et al., 2014), and positively affects spontaneous excitatory neurotransmission and drives synaptotagmin-7-dependent neurotransmitter release (Chanaday et al., 2021) – the aim of this study was to investigate expression and function of the alternative STIM2 splice variant STIM2.3 (also known as STIM2G).

## RESULTS

During previous initial screening for novel STIM2 splice variants (Miederer et al., 2015), we detected an alternatively spliced exon within the critical channel-activating region of STIM2 that leads to slightly longer STIM2.1 (NM_001169118.2) expressed in several cell types. However, we had been unable to detect any significant amount of the predicted variant STIM2.3 (NM_001169117.2; ENST00000467011.1) in human lymphocytes or cell lines. Meanwhile, detailed analysis of novel splice variants of STIM1 has revealed a new STIM1 variant with an alternative exon inserted between conventional exons 11 and 12, resulting in the short protein variant STIM1B, present in neuronal cells (Ramesh et al., 2021). Although the alternative *STIM1* exon is not per se conserved in *STIM2*, we find that the putative alternative *STIM2* exon 13 also resides within the intronic region between conventional exons 11 (now 12) and 12 (now 14), located on chromosome 4p15.2 at positions 27022623–27022690 (in the hg19 reference genome) or 27021001–27021068 (in the GRCh38/hg38 reference genome), with variant *STIM2.3* described according to the HGVS nomenclature as NC_000004.12(NM_020860.4): c.1763_1764 ins1764-1518_1764-1451, resulting in an mRNA

coding sequence of 2061 nucleotides, compared to 2502 nucleotides for *STIM2.2* (Fig. 1A). Of note, STIM2.2 contains an unusually long signal peptide, which starts at the MNAA sequence (residue 1, methionine; UniProt A0A8V8TMC8) and adds 87 amino acids to the pre-protein. This long and unconventional signal peptide is able to secrete a fusion protein and proves to have ER localization when the pre-sequence is present (Graham et al., 2011); however, most reports refer to Williams et al. (2001), where the non-AUG codon UUG (leucine 88) is identified as the start methionine (residue 1; UniProt Q9P246-3; see Fig. 1B). After translation, the inserted exon in STIM2.3 results in termination of the protein after 686 (numbering according to the long species; UniProt A0A8V8TMC8 with the long signal peptide region) or 599 amino acids (numbering according to UniProt Q9P246; short signal peptide region), and encodes a spliced-in 12-amino-acid sequence, which is unique to this splice variant. The abbreviated protein lacks 159 of the original C-terminal residues, including the serine/proline-rich (SP) region as well as the C-terminal PBD (Fig. 1B). In all following figures we will adhere to the amino acid numbering according to the shorter signal peptide, corresponding to UniProtKB/SwissProt IDs Q9P246-1 (STIM2.2) or H7C5A5 (STIM2.3). In contrast to previously analyzed variants (*STIM2.1, STIM1L, STIM1B, STIM1A*), this splice event apparently evolved more recently, and the inserted unique protein domain or nucleotide sequence so far could only be found in some subfamilies and species belonging to the parvorder Catarrhini, from which the superfamilies of Cercopithecoidea [see *Theropithecus gelada* (gelada baboons)] and Hominoidea derived ~32 million years ago, possibly implying a selective advantage (Fig. 1B,C). Of note, the annotation of the genomic region around the splice site may allow for an additional different upstream splice donor usage, which might lead to additional variants – but see also Maxeiner et al. (2023) for the pitfalls using sequence databases. In contrast to the alternative exon 16 (exon B) in *STIM1*, *STIM2* exon 13 also contains a short potential polyadenylation site.

Using conventional and splice-specific primers (Fig. 2A), as well as analyzing reads derived from publicly available RNA seq data (exemplary read counts shown in Fig. 2B), we detected *STIM2.3* in many, but not all, postmortem human brain probes extracted from different brain regions (Fig. 2C), with variability as expected for tissues derived from different postmortem times but potentially with a slightly higher expression of both *STIM2* and *STIM2.3* in female compared to male donors by reverse transcription–quantitative PCR (qRTPCR) analysis (Fig. 2C, red dots; also see below). In a more extensive analysis of exon expression data of STIM2 utilizing www.gtexportal.org, 20 exons are defined, with junctional reads to exon 2.3 (27021000–27021069, Chr 4) also found in skeletal muscle besides brain, and transcripts per million (TPM) reads for ENST00000467011.6 (STIM2.3) also seen in spleen, prostate and pituitary. However, in these tissues there is a clear discrepancy between isoform and junction-level quantification, which we also observed when analyzing additional datasets derived from human skeletal muscle and spleen (Fig. S1; also see Discussion). Focusing on expression in human brain, in which junctional and TPM reads coincide, we also used commercially available qRTPCR primers for general *STIM1* and *STIM2* detection and observed slightly increased *STIM1* versus *STIM2* expression in cerebellum (Fig. 2D). In addition, we also found *ORAI2* as the most abundant ORAI isoform in human cerebellum (Fig. 2D). We next quantified the fraction of exon 13 inclusion from the qRTPCR data, resulting in detection of a slightly increased abundance of exon inclusion in female donors (Fig. 2E). However, exon read analysis using published RNA seq data of postmortem tissues from additional NCBI Sequence Read Archive (SRA) sources, while confirming inclusion of exon 13 in

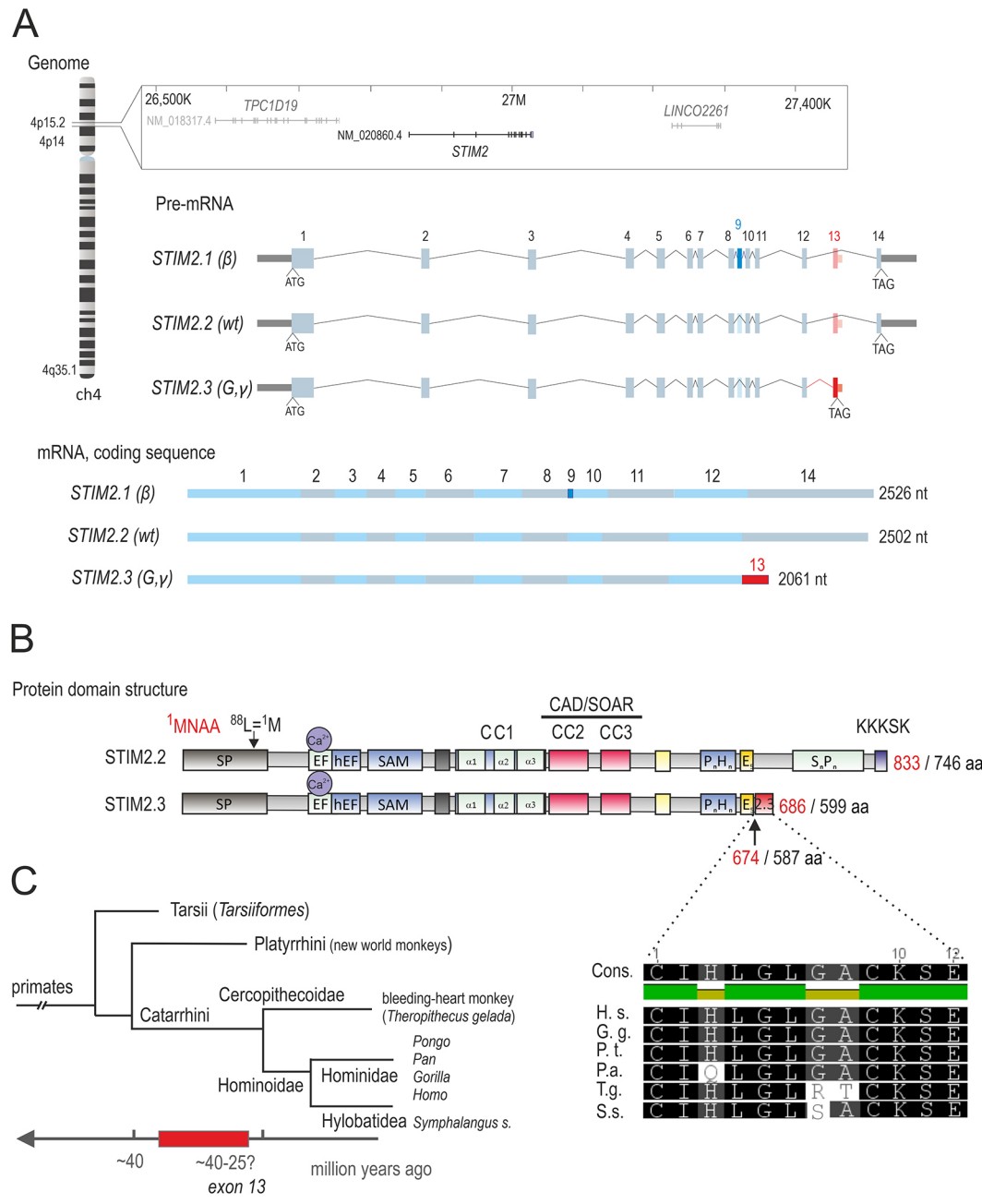

**Fig. 1. Human *STIM2* genomic context, conventional variants and phylogenetic tree of variant STIM2.3.** (A) Left and top: genomic context and gene structure of human *STIM2*. Middle: predicted pre-mRNA of the isoforms *STIM2.1* (β), *STIM2.2* (wt) and *STIM2.3* (G,γ). Translation start (ATG) and stop (TAG) codons are indicated. Conventional exons are depicted in light blue boxes, *STIM2.1*-specific exon (9) is highlighted in dark blue and *STIM2.3*-specific exon (13) is highlighted in red. Lines indicate intronic regions. Bottom: coding sequences of *STIM2.1*, *STIM2.2* and *STIM2.3* (nt, nucleotide). Alternating light blue and gray rectangles indicate individual exons. *STIM2.3*-specific exon (13) is highlighted in red. (B) Schematic protein structure of STIM2 displaying functional domains (aa, amino acid). The STIM2.3-specific domain (red) is inserted after a poly-glutamic acid domain ($E_5$, yellow) at amino acid 587 (Q9P246). SP, 1–13 (black text color); EF, 67–100; hEF, 101–131; SAM, 136–204; CC, CC1, 240–347; CAD, 348–452, with red numbers referring to UniProt A0A8V8TMC8 and black numbers to UniProt Q9P246. (C) Left: phylogenetic tree. Red bar indicates predicted evolutionary time line of the splice event of exon 13. Right: evolutionary conservation of the protein domain encoded by exon 13 in *Homo sapiens* (H. s.), *Gorilla gorilla* (G. g.), *Pan troglodytes* (P. t.), *Pongo abelii* (P.a.), *Theropithecus gelada* (T.g.) and *Symphalangus syndactylus* (S.s.). Identical residues are within black boxes. Cons, consensus sequence.

~18% of all reads (bulk RNA seq), did not reveal sex-specific differences (Fig. 2F). To estimate expression during early human brain development, we analyzed RNA seq data obtained from the human developmental biology resource (HDBR) database (Lindsay et al., 2016) to determine exon inclusion from brain RNA seq data obtained at ~7 (Carnegie stage 22) and 9 weeks after conception. We found that significantly less exon inclusion occurs in early fetal

development compared to postmortem tissue (Fig. 2G), suggesting that increased exon 13 inclusion might correlate with neuronal migration and synapse formation and pruning starting at ~9–12 gestational weeks. The developmentally late expression was also confirmed by the absence of *STIM2.3* expression ($2^{-\Delta Cq}$ of 3.3 µl template: 0.008±0.002; mean±s.e.m., *n*=4 independent donor cultures) in human stem cell-derived neuronal cultures (4–5 weeks

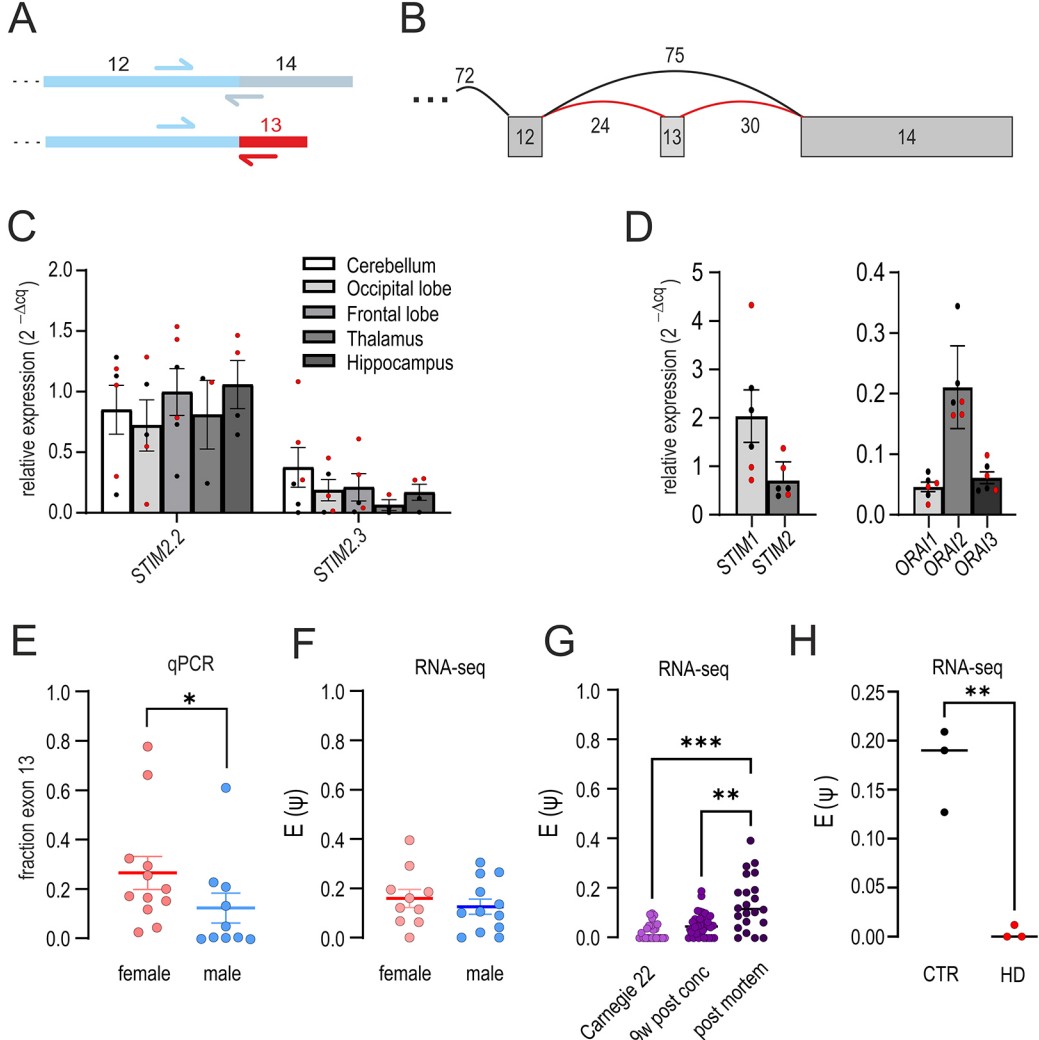

**Fig. 2. Expression analysis.** (A) Schematic representation of primer annealing sites on coding cDNA. Exon numbers are indicated. (B) Representative RNA seq read number indicating inclusion of exon 13 (red) and exclusion of exon 13 (black), derived from postmortem human brain sequence datasets described in the Materials and Methods. (C) Relative expression (mean $2^{-(\Delta Cq)}$ ±s.e.m.) of STIM2.2 and STIM2.3 detected using splice-specific primers in different postmortem human brain regions, as indicated, derived from six different donors. Male donors are shown in black, female donors are shown in red. (D) Relative expression (mean $2^{-(\Delta Cq)}$±s.e.m.) of STIM1, STIM2 and ORAI1–ORAI3 in cDNA of postmortem human cerebellum derived from six different donors. Male donors are shown in black, female donors are shown in red. (E) Quantification of fraction exon 13 as determined by qRTPCR normalized to sum of splice-specific and wild-type-specific STIM2 measured in D from postmortem female and male donors. Mean±s.e.m. *$P<0.05$ (Mann–Whitney test). (F) Aggregate percent splice Inclusion [PSI, E(Ψ)] in female and male postmortem donors using RNA seq data of cerebellum and frontal cortex. Mean±s.e.m. (G) Aggregate PSI of different human developmental stages utilizing RNA seq data of cerebellum, cortex and telencephalon [Carnegie stage 22 (~52–55 days post fertilization), $n=23$; 9w post conc, 9 weeks post-conception ($n=22$)]. Lines indicate the mean values. **$P<0.01$, ***$P<0.01$ (Kruskal–Wallis ANOVA). (H) PSI in samples from postmortem control (CTR) and HD patients, $n=25$, extracted from Elorza et al. (2021). Lines indicate median values. **$P<0.01$ (unpaired $t$-test).

of differentiation; see Rizo et al., 2022), in contrast to STIM2.2 expression ($2^{-\Delta Cq}$ of 0.5 µl template: 0.35±0.08; mean±s.e.m., same donor cDNA), making physiological investigations of STIM2.3 in these cultures not possible. We also discovered STIM2 splicing information in a dataset published on mis-splicing in Huntington's disease (HD), wherein postmortem brain regions of HD patients were compared to those of healthy controls (Elorza et al., 2021). Within this dataset, STIM2 exon13 inclusion is significantly reduced in HD patients (Fig. 2H). Although glial cells contain relatively high expression of STIM2, we did not detect splice inclusion in samples (0/4) derived from astrocytes differentiated from human induced pluripotent stem cells (data not shown), indicating that STIM2.3 splicing likely is more prominent in neurons. This is also consistent with analysis of primary

glioblastoma cDNA samples, with STIM2.3-specific PCR yielding $2^{-\Delta Cq}$ values of 0.14±0.08 (mean±s.d., $n=3$) for neural-derived glioblastoma versus 0.02±0.02 (mean±s.d., $n=3$) for mesenchymal-derived glioblastoma or 0.004±0.0025 (mean±s.d., $n=3$) for classic glioblastoma. For further confirmation of lineage-specific splicing regulation of STIM2.3 in the primate brain, see also table S1 of Recinos et al. (2024).

**STIM2.3 increases store-operated Ca²⁺ entry**

To assess functional differences of the alternative STIM2 protein variants, we first expressed YFP-tagged (downstream of the SP region) isoforms in HEK293 (hereafter HEK) cells lacking endogenous levels of STIM1 and STIM2 (Zhou et al., 2018). Whereas cells transfected with an empty control vector displayed

low resting Ca²⁺ levels and no store-operated re-entry of Ca²⁺ after thapsigargin (TG)-induced store depletion, expression of wild-type STIM2.2 in the absence of STIM1 only slightly raised basal Ca²⁺ levels and recovered a small re-entry after re-addition of Ca²⁺ to the medium (Fig. 3A). Surprisingly, the C-terminally deleted variant STIM2.3, while also showing a higher basal Ca²⁺ level compared to controls, but similar to STIM2.2, showed increased rates, peak and

plateau of SOCE (Fig. 3A,B). To validate these results in a cell line more closely resembling neurons and with endogenous levels of STIM1 present, we generated SH-SY5Y cells lacking only STIM2 by CRISPR/Cas9, demonstrating a significant reduction of SOCE (Fig. S2A,B). Compared to HEK cells, SH-SY5Y cells show higher expression of endogenous *ORAI2* (Fig. S2C). Re-expression of either STIM2 variant in these cells and calibration of the Fura-2

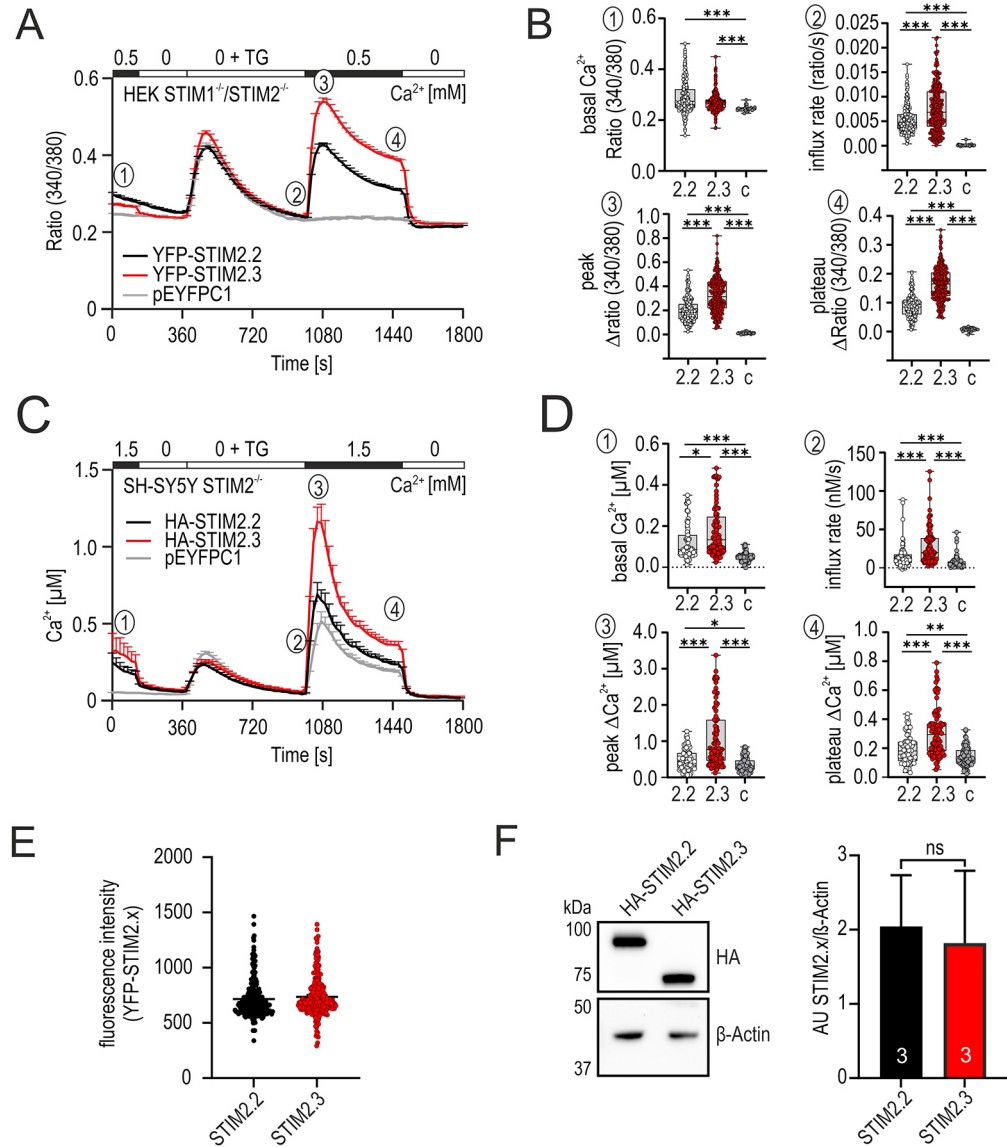

**Fig. 3. STIM2.3 increases SOCE compared to STIM2.2.** (A) Traces showing average changes (mean+s.e.m.) in intracellular Ca²⁺ (ratio 340/380) over time in response to perfusion of different external Ca²⁺ concentration (in mM) and TG (1 μM), as indicated in the upper bar, in HEK STIM1/2 DKO cells transfected with STIM2.2 (black, *n*=227), STIM2.3 (red, *n*=232) or vector only control (pEYFPC1, gray, *n*=27). Numbers indicate features quantified in B. (B) Quantification of changes in ratio of resting (basal Ca²⁺), influx rate (ratio/s), Δpeak and Δplateau measured in panel A (c, vector-only control). Data are represented as box and whisker plots. The box marks the 25th to 75th percentiles, with the median indicated by a line. Whiskers show the minimum to maximum range, with all points indicated. *n* as per A. ***P<0.001; Kruskal–Wallis ANOVA with Dunn's multiple comparisons test. (C) Calibrated traces showing average changes (mean+s.e.m.) in intracellular Ca²⁺ concentration (in μM) over time in response to perfusion of different external Ca²⁺ concentration (in mM) and TG (1 μM), as indicated in the upper bar, in SH-SY5Y STIM2⁻/⁻ cells after transfection with STIM2.2 (black, *n*=85), STIM2.3 (red, *n*=89) or vector only (pEYFPC1, gray, *n*=94). Numbers indicate cells quantified in D. (D) Quantification of changes in Ca²⁺ concentration of resting (basal Ca²⁺), influx rate (nM/s), Δpeak and Δplateau measured in panel C (c, vector-only control). Data are represented as box and whisker plots. The box marks the 25th to 75th percentiles, with the median indicated by a line. Whiskers show the minimum to maximum range, with all points indicated after removal of outliers (ROUT method) to provide appropriate scaling with no change in the statistical power compared to values before removal. *P<0.05, **P<0.01, ***P<0.01; Kruskal–Wallis ANOVA with Dunn's multiple comparisons test. (E) Quantification of YFP fluorescence intensity from cells measured in panel A (arbitrary units, AU). Line marks the median. *n*=227 (STIM2.2), *n*=232 (STIM2.3) as in A. (F) Western blot showing HA–STIM2.2 and HA–STIM2.3 levels following heterologous expression in HEK STIM1/2⁻/⁻ cells; quantification relative to β-actin levels from three independent transfections (mean±s.d.; AU, arbitrary units). ns, not significant; unpaired, two-tailed *t*-test.

Journal of Cell Science

ratios to yield absolute $Ca^{2+}$ concentration values uncovered a small further increased basal $Ca^{2+}$ concentration upon STIM2.3 expression when compared to STIM2.2, whereas both showed increased basal $Ca^{2+}$ concentration compared to vector-transfected cells (Fig. 3D), and otherwise confirmed the phenotype of increased SOCE seen in the STIM1 STIM2 double-knockout (STIM1/2 DKO) HEK background (Fig. 3C,D). As expected with endogenous levels of STIM1 present, SOCE in the absence of STIM2 (gray trace) was not eliminated, and STIM2.2 re-expression only slightly increased SOCE (Fig. 3C,D). Equal STIM2 variant expression was controlled by recording the tagged YFP fluorescence of measured cells (Fig. 3E) and, in addition, by determining protein expression of constructs in which the N-terminal YFP was replaced by an HA tag, yielding an expected molecular mass of ∼68 kDa for STIM2.3 (Fig. 3F). Although the relative ratios of alternative STIM2 variants in a given single human neuron are unknown, we checked whether STIM2.3 also increased SOCE in the presence of STIM2.2. Co-expression of YFP-tagged STIM2.2 or STIM2.3 with mKate2-tagged STIM2.2 or STIM2.3 demonstrated that the presence of STIM2.3 has a reduced but significant amplifying effect on rate, peak and plateau of SOCE (Fig. S2D,E), while showing no dominant effect on basal $Ca^{2+}$ in the absence of STIM1 (Fig, S2E). Interaction analysis utilizing bimolecular fluorescence complementation assays (BiFC) confirmed that STIM2.3 interacts with either STIM1 or STIM2.2, and also shows an increased interaction with ORAI2 when compared with STIM2.2 (Fig. S2F).

### Increased function does not require splice-specific residues and is not phenocopied by deletion of both EB-binding sites and the PBD

As the short STIM1 variant STIM1B reduces function in part in a sequence-specific manner (Ramesh et al., 2021) and to address whether the gain-of-function phenotype observed for STIM2.3 is related to its specific spliced-in residues, we created a deletion mutant that terminates the wild-type STIM2 protein after residue 587 (STIM2.2Δ587), the last common residue of both variants. In addition, we recreated a deletion of only the last five C-terminal lysine residues (STIM2.2Δ5K) – disrupting the C-terminal phospholipid-binding PBD, which has previously been shown to result in a strong loss-of-function phenotype (Ong et al., 2015) and is a necessary component for the STIM1–NFAT signaling complex (Son et al., 2020) – as well as additional variants described below, schematically depicted in Fig. 4A. Indeed, when comparing SOCE signatures of these constructs in HEK STIM1/2 DKO cells, we found that deletion of only the PBD (STIM2.2Δ5K) reduced basal $Ca^{2+}$ levels and SOCE to a significant extent, whereas the deletion after residue 587 phenocopied STIM2.3, suggesting that deletion of an additional C-terminal inhibitory domain/binding site, and not inclusion of splice-specific residues, leads to the STIM2.3 gain-of-function phenotype (Fig. 4B,C). To investigate whether retention of STIM2.2Δ5K at the microtubule network via its potential dual microtubular EB-binding motifs ([686]SGIP[689] and [718]SSIP[721]) is one reason for its strong loss-of-function phenotype, and to test whether the lack of these motifs in STIM2.3 is sufficient to explain its gain-of-function phenotype, we deleted both motifs in the background of STIM2.2 and STIM2.2Δ5K (Fig. 4A; STIM2.2 2×IP and STIM2.2 2×IP+Δ5K, respectively). Whereas deletion of the EB-binding motifs in the STIM2.2Δ5K mutant indeed rescued its reduced SOCE phenotype back to wild-type levels, STIM2.3 still showed a significantly increased SOCE in comparison (Fig. 4D,E). Interestingly, an additional deletion mutant, creating a slightly longer STIM2 compared to STIM2.3 (STIM2.2Δ624) did not

phenocopy STIM2.3 or STIM2.2Δ587 but showed wild-type-equivalent SOCE levels (Fig. S3), indicating a potential novel negative regulatory site between residues 587 and 624 (37 amino acids) within STIM2. However, deletion of only this region from full-length STIM2 reduced SOCE instead of increasing it (Fig. S3; see also Discussion).

### Localization and cluster formation

Expression of STIM2.2 has been shown to lead to pre-clustered and pre-activated SOCE complexes (Brandman et al., 2007; Miederer et al., 2015; Ong et al., 2015). We used C-terminally mCherry-tagged STIM1 or STIM2.2 with co-expression of N-terminally YFP-tagged STIM2.3 to assess the degree of colocalization and pre-clustering of the isoforms. In the absence of stimulation, STIM2.3 showed good colocalization with either STIM1 or STIM2 in ER regions close to the nucleus; however, STIM2.3 was significantly less pre-clustered with STIM2.2 in regions near the PM at rest (Fig. 5A,B). The reduced PM pre-clustering of STIM2.3 could be phenocopied either by deletion of the PBD alone or by deletion at residue 587, with STIM2.2Δ587 mimicking the SOCE phenotype of STIM2.3, thus demonstrating that the PBD alone stabilizes formation of pre-clusters at the PM (Fig. S4A). Strong TG-induced store depletion abolished differences in colocalization (Fig. 5C,D), and no differences concerning co-cluster sizes could be detected when comparing STIM2.3 co-clustering with STIM1 or STIM2.2 (Fig. 5E).

We next asked whether the degree of TG-induced cluster formation and the cluster size depended on the presence or absence of either the phospholipid-binding PBD, the microtubular attachment EB-binding sites or on the larger deletion found in STIM2.3. Cells were transfected with YFP-tagged constructs and PH-PLC–mCherry to clearly mark the plane of the PM for total internal reflection fluorescence (TIRF) microscopy and to measure regions of high phosphatidylinositol (4,5)-bisphosphate ($PIP_2$), and were then imaged before and after addition of TG (representative images shown in Fig. 6A). Despite the ability of STIM2.3 to increase SOCE, STIM2.3, STIM2.2Δ5K and STIM2.2 2×IP+Δ5K all showed a significant reduction in mean cluster sizes and cluster intensities (Fig. 6B,C), demonstrating that cluster size is not a valid predictor of SOCE size (Fig. 4B–E). In addition, as recently shown (Cohen et al., 2023; Gulyas et al., 2022), phosphatidylinositol 4-phosphate (PI4P) and not $PIP_2$ critically regulates STIM1 and STIM1 CAD/SOAR attachment to the PM, which partially explains why areas of high PH-PLC–mCherry (indicating high $PIP_2$; Fig. 6A) did not fully overlap with clusters of the STIM2 fusion proteins. Of note, these experiments were performed in the absence of STIM1. Analyzing the frequency distribution of cluster sizes (Fig. 6C), it became apparent that only full-length STIM2.2 is able to efficiently form cluster sizes greater than ∼0.7 $\mu m^2$, confirming a role of the PBD in cluster size expansion and probably in cluster size stability.

### Splice isoform-dependent effects on NFAT activation

To assess downstream effectors of SOCE, we investigated the ratio of cytosolic to nuclear GFP-tagged NFATc1 in STIM1/2 DKO SHSY-5Y cells. mKate2-tagged STIM2 constructs were co-transfected with GFP-tagged NFATc1 and the nuclear to cytosolic ratios of NFATc1 were analyzed before and after stimulation. As expected from its higher degree of PM pre-clustered protein with the presence of its PBD, expression of STIM2.2, but not of STIM2.3 or deletion mutants without a PBD, showed a high degree of basal NFATc1 translocation (Fig. 7A,B) in the absence of stimulation. Whereas store depletion did not lead to additional translocation in

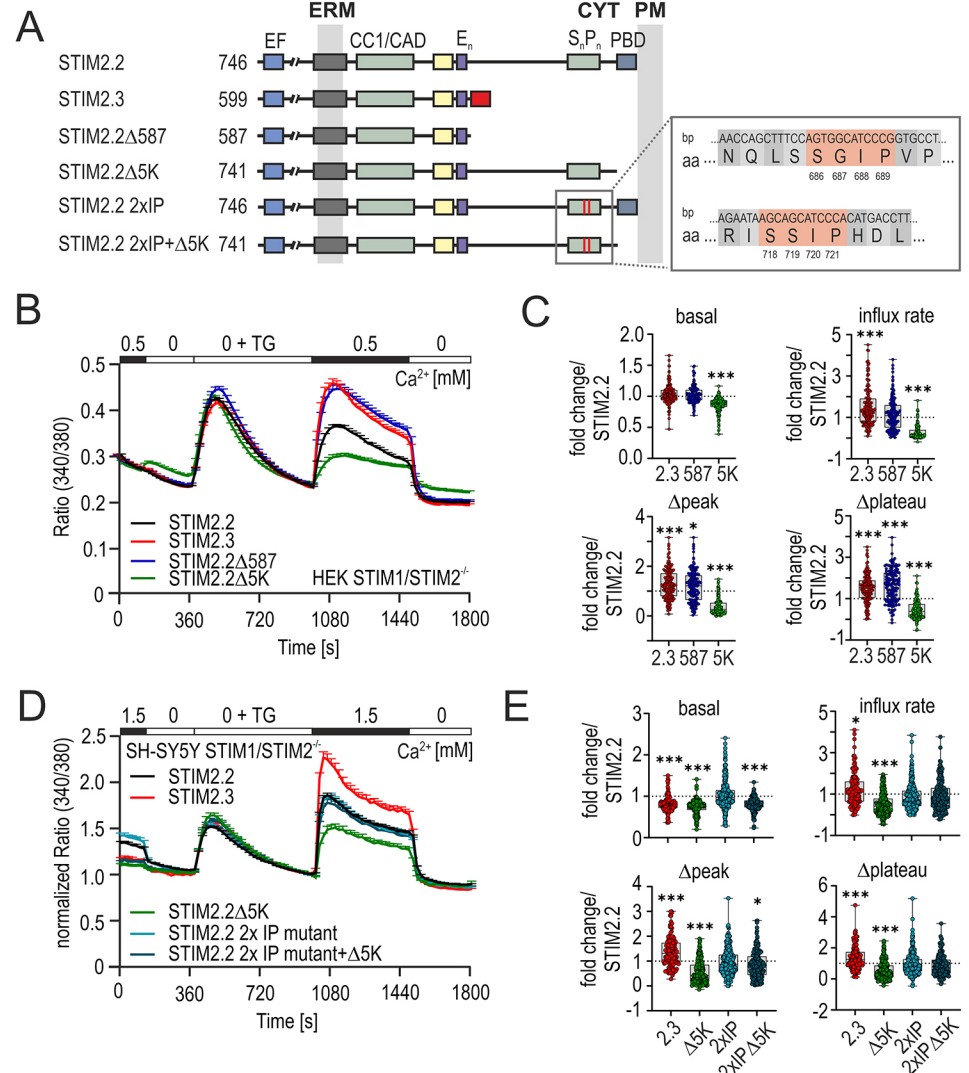

**Fig. 4. Structure–function analysis of C-terminal domains.** (A) Schematic protein structure displaying functional domains of STIM2 splice variants and deletion mutants (amino acid numbering as in Fig. 1, for STIM2.2Δ587, STIM2.2Δ5K, STIM2.2 2×IP and STIM2.2 2×IPΔ5K). EB-binding motifs are indicated as red lines within the $S_n/P_n$ rich region. Corresponding sequence [base pair (bp) and amino acid (aa)] information is displayed in the gray box (right) with relevant amino acids indicated. CC1/CAD, CC1 + CAD, aa 240–452; CYT, cytoplasm; ERM, ER membrane. (B) Traces showing average changes (mean+s.e.m.) in intracellular $Ca^{2+}$ (ratio 340/380) over time in response to perfusion of different external $Ca^{2+}$ concentrations (in mM) and TG (1 µM), as indicated in the upper bar, in HEK STIM1/2 DKO cells transfected with STIM2.2 (black, $n=144$), STIM2.3 (red, $n=145$), STIM2.2Δ587 (blue, $n=132$) or STIM2.2Δ5K (green, $n=164$). (C) Quantification of changes in ratio of basal, influx rate (ratio/s), Δpeak and Δplateau as fold change relative to STIM2.2, as measured in panel B. Data are represented as box and whisker plots. The box marks the 25th to 75th percentiles, with the median indicated by a line. Whiskers show the minimum to maximum range, with all points indicated. $n$-values as in B. *$P<0.05$, ***$P<0.001$; Kruskal–Wallis ANOVA with three-way multiple comparisons. (D) Normalized average traces showing changes (mean+s.e.m.) in intracellular $Ca^{2+}$ (ratio 340/380) over time in response to perfusion of different external $Ca^{2+}$ concentrations (in mM) and TG (1 µM), as indicated in the upper bar, in SH-SY5Y STIM1/2$^{-/-}$ cells transfected with YFP–STIM2.2 (black, $n=198$), YFP–STIM2.3 (red, $n=130$), YFP–STIM2.2Δ5K (green, $n=214$), YFP–STIM2.2 2×IP (light blue, $n=158$) or YFP–STIM2.2 2×IPΔ5K (petrol blue, $n=168$). (E) Quantification of changes in resting $Ca^{2+}$ (basal), influx rate, Δpeak and Δplateau measured in panel D as fold change normalized to STIM2.2. Data are represented as box and whisker plots. The box marks the 25th to 75th percentiles, with the median indicated by a line. Whiskers show the minimum to maximum range, with all points indicated. $n$ as indicated in D. *$P<0.05$, **$P<0.01$, ***$P<0.001$; Kruskal–Wallis ANOVA with multitple comparisons against STIM2.2.

the case of STIM2.2, STIM2.3 and the STIM2.2 2×IP+Δ5K mutant showed significant changes in NFATc1 translocation ratios upon stimulation, despite their lack of the PBD (Fig. 7B). Here, it also appeared that a tug-of-war between PM attachment via the PBD and retention to the microtubules via the EB-binding IP motifs also reflects the ability to initiate NFATc1 translocation, even in the absence of STIM1. Although STIM2.2Δ5K was unable to cause NFATc1 translocation, in accordance with previously reported results (Son et al., 2020), additional mutations within the IP motifs were sufficient to restore SOCE (Figs 4D,E and 7E) as well as to

recover NFATc1 translocation, with no significant differences in the TG-induced translocation between STIM2.2, STIM2.3 and STIM2.2 2×IP+Δ5K (Fig. 7A,B), confirming the submembrane microdomain as an essential docking site to initiate NFATc1 translocation (Subedi et al., 2018) and in line with the AKAP-binding domain on ORAI1, which anchors NFATc1 to the SOCE complex (Kar et al., 2021). To correlate global changes in intracellular $Ca^{2+}$ concentration and NFATc1 translocation in a comparable manner, we also recorded SOCE with the same time course and parameters as used for the NFATc1 translocation assays

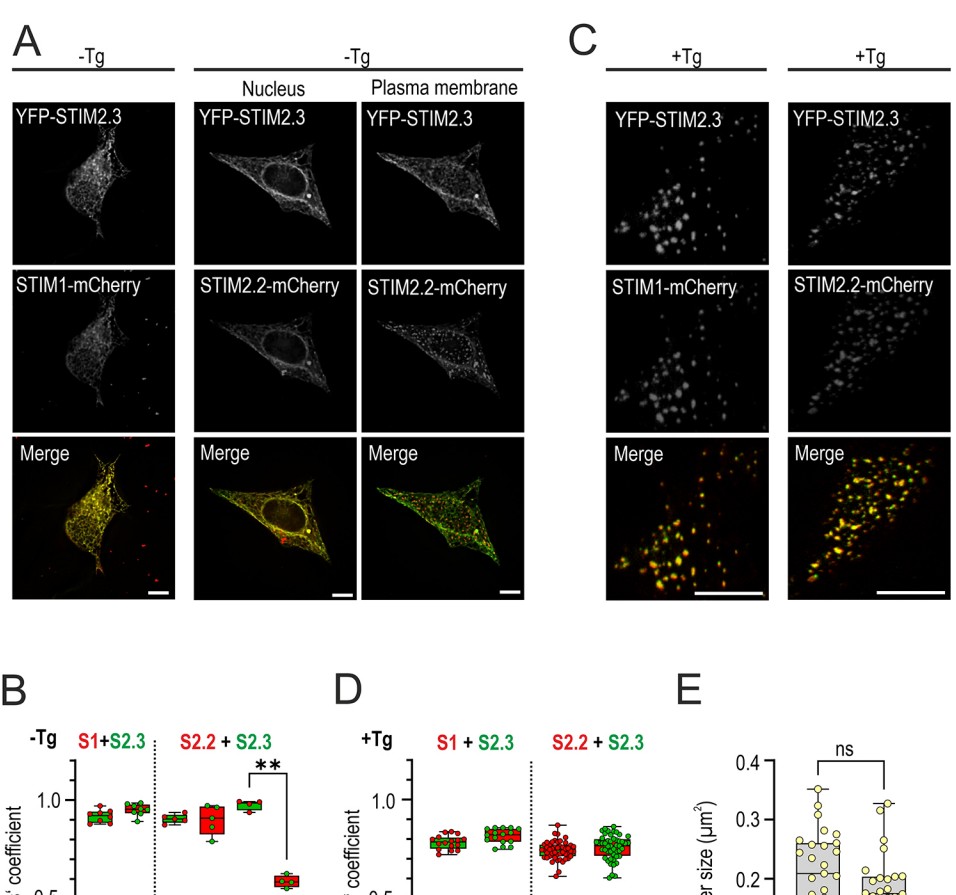

**Fig. 5. Localization and cluster formation.** (A,C) Representative images of HEK STIM1/2 DKO cells co-transfected with YFP–STIM2.3 (grayscale, 488 nm) and either STIM1–mCherry (grayscale, 560 nm; left) or STIM2.2–mCherry (grayscale, 560 nm; right) before (A) and after (C) stimulation with TG (1 µM) for 20 min. Merge images show the YFP signal in green and the mCherry signal in red. Scale bars: 10 µm. (B,D) Colocalization analysis of cells from A and C, respectively. For each condition 5–45 cells from three independent transfections were analyzed using Manders' overlapping coefficients (M1, M2). The color of each dot (cell) within the colored box plot represents the Manders' coefficient of the constructs written above. Nucleus refers to the focus plane of the nucleus, PM to the focus plane of the PM. For D and E (+TG), all analysis is done at the PM focus plane. **$P<0.01$; Kurska–Wallis ANOVA with six-way comparisons. (E) Quantification of cluster sizes from cells measured in C. Each dot represents the cluster size average of single cell. For each condition 16–45 cells from three independent transfections were analyzed. Unpaired two-tailed $t$-test of normal distributed data revealed no significance (ns). Data in B,D,E are represented as box and whisker plots. The box marks the 25th to 75th percentiles, with the median indicated by a line. Whiskers show the minimum to maximum range, with all points indicated.

and quantified resting $Ca^{2+}$ and total stimulated $Ca^{2+}$ entry as area under the curve (AUC) during stimulation. Fig. 7C shows mean global $Ca^{2+}$ responses to a 30 min stimulation. Whereas vector-only and STIM2.2Δ5K transfection showed neither basal $Ca^{2+}$ entry nor a sustained SOCE and did not enable NFATc1 translocation, the other constructs reached a critical sustained intracellular $Ca^{2+}$ concentration threshold to initiate nuclear translocation. These results also demonstrate the non-linearity between SOCE and endpoint (30 min) NFATc1 translocation, as despite different degrees of $Ca^{2+}$ entry (Fig. 7D), STIM2.2, STIM2.3 and STIM2.2 2×IP+Δ5K all resulted in a similar degree of translocation 30 min after stimulation (Fig. 7B). The finding that STIM2.3 or the STIM2.2 2×IP+Δ5K mutant – despite also raising basal intracellular $Ca^{2+}$ concentration – did not lead to pre-stimulus-translocated NFATc1 is congruent with their significantly decreased degree of ER–PM junction pre-clustering (Fig. 6) and implies that pre-cluster size correlates with an increased recruitment and activation of the ORAI1–AKAP–NFAT signalosome.

## STIM2.3 has a reduced interaction with AMPK compared to STIM2.2

AMP-activated protein kinase (AMPK) is a master regulator of cellular energy metabolism as it senses and monitors the concentration of the nucleosides AMP, ADP and ATP, which all bind competitively to the γ

subunit of the heterotrimeric αβγ complex of the active enzyme. Upon activation and autophosphorylation either by liver kinase B1 (LKB1, also known as STK11; canonical pathway) in conjunction with altered ATP/AMP ratios or by $Ca^{2+}$/calmodulin-dependent protein kinase kinase 2 (CaMKK2) as a non-canonical and energy-independent AMPK activation pathway, active AMPK upregulates catabolic pathways while inhibiting anabolic pathways. SOCE is one source of $Ca^{2+}$ entry necessary for CaMKK2 activation (Chauhan et al., 2019), and both STIM1 and STIM2 are AMPK phosphorylation substrates (Stein et al., 2019). STIM2.2 also functions as an AMPK scaffolding protein, promoting assembly and activation of a trimeric complex between STIM2, AMPK and CaMKK2 (Chauhan et al., 2019). Within STIM2.2, the C-terminal region spanning amino acids 442–746 has been identified as the interacting domain with AMPK, with a STIM2.2 deletion construct (amino acids 1–441) failing to interact with AMPK (Chauhan et al., 2019). Thus, we asked whether the truncated C terminus of STIM2.3 (last common amino acid 587) interfered with this interaction. Co-immunoprecipitation experiments in HEK STIM1/2 DKO cells transfected with HA–STIM2.2 or HA–STIM2.3 showed a strong interaction of endogenous AMPK with STIM2.2 and revealed a remaining, but reduced, interaction of STIM2.3 with the catalytic subunit AMPKα (sample blot shown in Fig. 7F and quantification shown in Fig. 7G, back circles, with data derived from several independent experiments; see also Fig. S6, blot

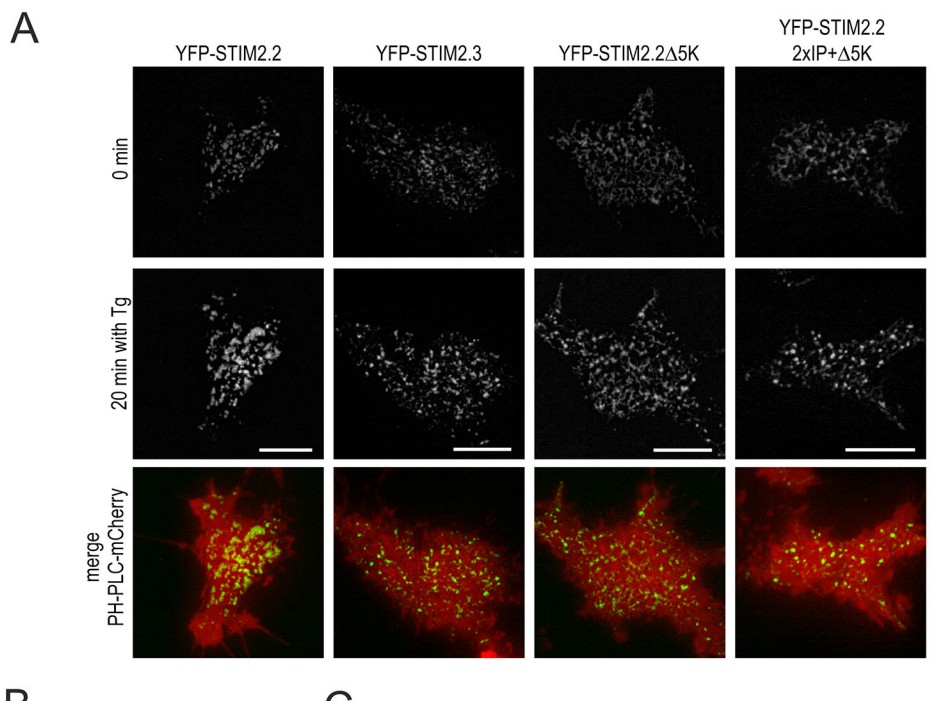

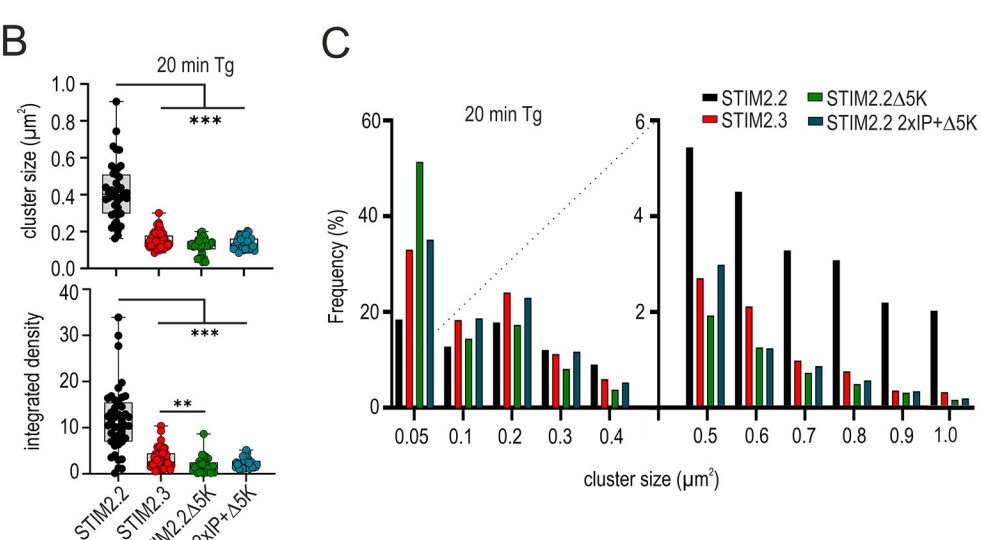

**Fig. 6. Cluster size distribution of STIM2 variants.** (A) Representative images of HEK STIM1/2 DKO cells co-transfected with PH-PLC–mCherry (red, merge) and YFP–STIM2.2, YFP–STIM2.3, YFP–STIM2.2Δ5K or YFP–STIM2.2 2×IP+Δ5K before (0 min) and after (20 min) stimulation with 1 μM TG. Scale bars: 10 μm. (B) Quantification of cluster size and integrated density measured from cells as in A. For each condition, 34–50 cells from three independent transfections were analyzed. Data are represented as box and whisker plots. The box marks the 25th to 75th percentiles, with the median indicated by a line. Whiskers show the minimum to maximum range, with all points indicated. **$P<0.01$, ***$P<0.001$; Kruskal–Wallis ANOVA with four-way comparsions (only significant differences are shown). (C) Cluster size distribution (frequency) measured from cells as in A,B.

transparency). No interaction was found with HA-tagged full length STIM1 or a deletion construct abbreviating STIM1 before the CAD domain (Fig. S4B). Transient expression of the variants also had no significant effect on endogenous AMPK levels (Fig. S4C). Regarding STIM2.2 and STIM2.3, we did not find a significant influence of store depletion (with or without 10 min TG treatment) on this interaction (Fig. 7F, lower panel; quantification in Fig. 7G, gray circles). Additionally, to test whether isoform-induced altered $Ca^{2+}$ fluxes before or after TG treatment differentially affected activation of bound AMPK, potentially through CaMKK2, we investigated the phosphorylation state of AMPKα at Thr172 within its activation loop by probing with a Thr172-phosphospecific AMPK antibody. Concomitant with the reduced interaction of STIM2.3, we found reduced amounts of phospho-AMPK bound to STIM2.3 as compared to that bound to STIM2.2 (Fig. 7F,H). In STIM1/2 DKO HEK cells with re-expression of STIM2 variants, we only resolve endogenous phospho-AMPK in the STIM2-bound fractions and not within the lysate fractions, preventing an analysis of the endogenous activation

state of unbound AMPK in dependence of STIM2 variant expression, and quantification of bead-bound phospho-AMPK relative to bead-bound AMPK would factor out any effect due to reduced interaction. To conclude, our results show a reduced interaction of STIM2.3 with AMPK compared to the full-length STIM2 protein, resulting in a correspondingly reduced AMPK activation state, despite an increased TG-induced SOCE.

**STIM2.3 in murine neurons neither shows preferential localization to nor alters the number of synapses, but increases dendritic spine size in a splice-specific manner**

The short neuronal-specific STIM1 variant (STIM1B) shows increased localization to presynaptic ER when compared to the full-length isoform and leads to enhanced synaptic release upon high frequency stimulation (Ramesh et al., 2021). To investigate localization of STIM2.3 versus STIM2.2, we transduced primary hippocampal neurons with tagged variants under the control of a CaMKIIα promoter to drive expression in neurons of both

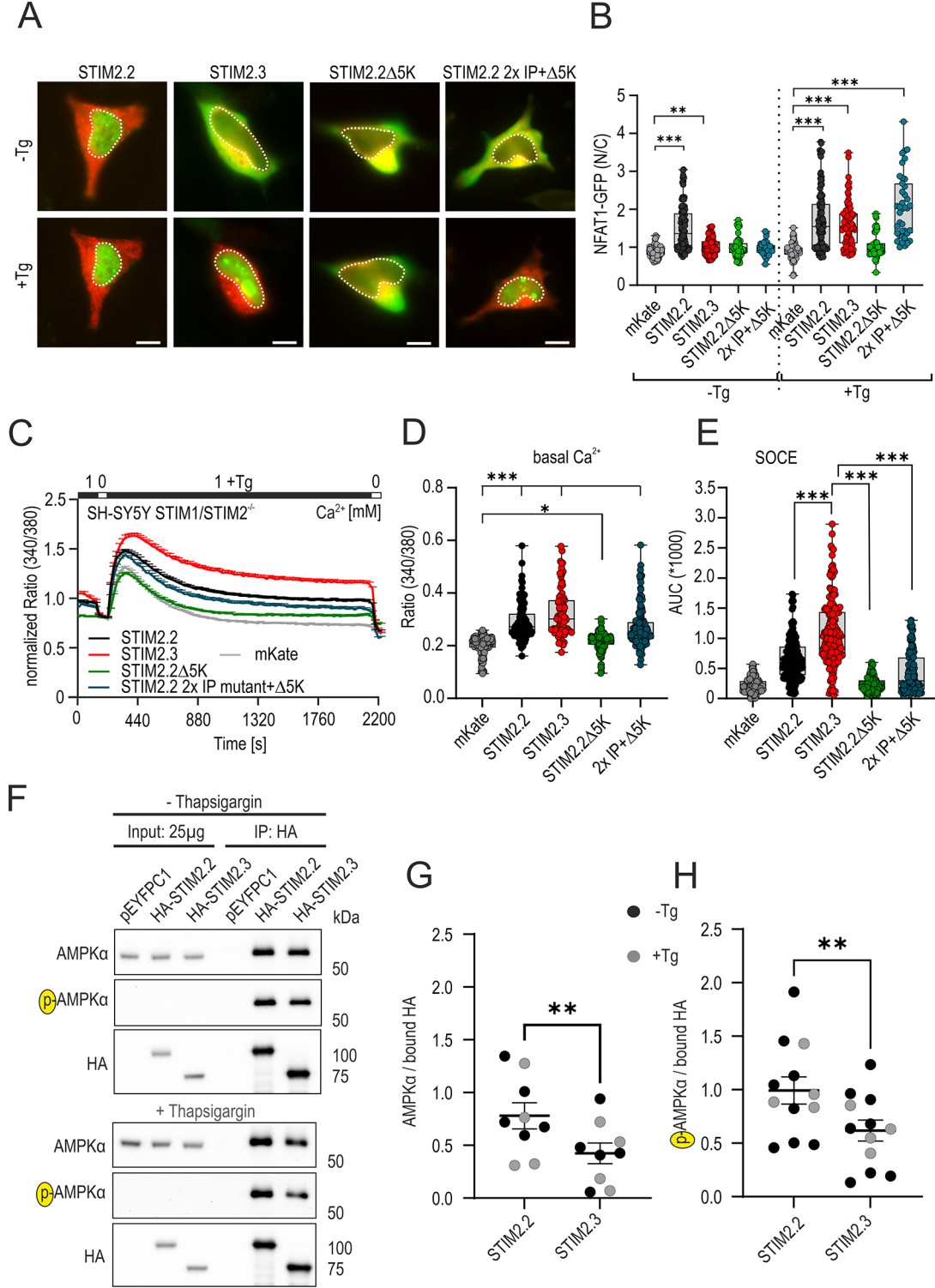

**Fig. 7.** See next page for legend.

mass cultures or autaptic cultures derived from C57BL/6N mice. Overall, using immunohistochemistry, we found either variant within MAP2-positive dendrites and to a reduced degree in NFL (NEFL)-positive axons (Fig. S5A), but with no differential effect on mean dendrite length (Fig. S5B). Using mass cultures, we also did not detect differences in colocalization between either STIM2.2 or STIM2.3 and the presynaptic marker protein synapsin or with the

postsynaptic marker protein PSD95 (also known as DLG4) (Fig. 8A). As quantification of the number of pre- or post-synaptic specializations is difficult in mass cultures, we made use of autaptic cultures, in which hippocampal neurons were seeded at low density (1000 cells/ml) onto a layer of glial micro-islands, resulting in single neurons making autaptic synapses onto themselves (autapses; Bekkers, 2020), which allows counting of synapses

**Fig. 7. STIM2 splice isoform effects on basal and induced NFATc1 activation and on interaction with AMPK.** (A) Representative images of SH-SY5Y STIM1/2$^{-/-}$ cells co-transfected with NFATc1–GFP (green) and either mKate2–STIM2.2 (red), mKate2–STIM2.3 (red), mKate2–STIM2.2Δ5K (red) or mKate2–STIM2.2 2×IP+Δ5K (red) before (−Tg) and after (+Tg) stimulation with 1 μM TG for 30 min. Dotted lines mark the nucleus. Scale bars: 10 μm. (B) Changes in nuclear/cytosolic (N/C) ratio of NFATc1-GFP before and after stimulation with TG in cells transfected as indicated. For each condition 39–114 cells were analyzed from three independent transfections. Data are represented as box and whisker plots. The box marks the 25th to 75th percentiles, with the median indicated by a line. Whiskers show the minimum to maximum range, with data points indicated. **$P<0.01$, ***$P<0.001$; Kruskal–Wallis ANOVA with four-way comparison against control transfected cells. No statistical difference in comparison of NFATc1 translocation after TG between STIM2.2, STIM2.3 and STIM2.2 2×IP+Δ5K. For reasonable scaling, outliers were removed after ROUT analysis with no change in statistical significances. (C) Normalized traces showing global changes (mean+s.e.m.) in intracellular Ca$^{2+}$ (ratio 340/380) after perfusion of different external Ca$^{2+}$ concentrations (in mM) and TG (1 μM), as indicated in the upper bar, in SH-SY5Y STIM1/2$^{-/-}$ cells after transfection with mKate2–STIM2.2 (black, n=184), mKate2–STIM2.3 (red, n=160), mKate2–STIM2.2Δ5K (green, n=156), mKate2–STIM2.2 2×IP+Δ5K (dark blue, n=164) or mKate vector only (gray, n=241). (D) Quantification of resting Ca$^{2+}$ (ratio 340/380, not normalized) for cells as in C. *$P<0.05$, ***$P<0.001$; Kruskal–Wallis ANOVA with five-way multiple comparisons. (E) Quantification of AUC during SOCE from cells measured in C. ***$P<0.001$; Kruskal–Wallis ANOVA with five-way multiple comparisons. Not all signifances are shown for better readability; with the exception of comparing mKate control with mKate2–STIM2.2Δ5K (not significant), all other pairwise comparisons are highly significant (***$P<0.001$). Data in D and E are represented as box and whisker plots. The box marks the 25th to 75th percentiles, with the median indicated by a line. Whiskers show the minimum to maximum range, with points indicated. For reasonable scaling in E, outliers were removed after ROUT analysis with no change in statistical significances. (F) Representative western blots showing immunoprecipitation (IP) of HA–STIM2.2 or HA–STIM2.3 transfected constructs with endogenous AMPKα in HEK STIM1/2 DKO cells before (upper blot) and after (lower blot) stimulation with 1 μM TG for 10 min, using anti-HA agarose. Membranes were incubated with the indicated antibodies. Cells expressing pEYFPC1 were used a control (but see also Fig. S4B). P-AMPKα, Thr172 phospho-AMPK. (G,H) Densitometric quantification of bound AMPKα and P-AMPKα signals before and after TG treatment measured from eight (before TG) and four (with TG) independent transfections. Mean±s.e.m. is indicated. (G) AMPKα signal was normalized to non-saturated HA-bound bead signal, and normalized signals were compared using a non-parametric two-tailed paired t-test, where samples of the same blot are paired. (H) Corresponding normalization of the phosphorylation-specific AMPK antibody to HA input. Statistical test as in G. Additional blots from which quantifications were derived are shown in Fig. S6 (blot transparency). **$P<0.01$.

from a single neuron. However, we found differences in neither the number of synapsin-positive presynaptic varicosities, nor the number of PSD95-positive postsynaptic specializations upon overexpression of STIM2.3 when compared to STIM2.2 (Fig. 8B). Given the role of STIM2 role in shaping spine morphology (Popugaeva et al., 2015), we set out to investigate whether STIM2.2 or STIM2.3 expression had differential effects on spine size. Transfections of either variant together with mGreenLantern, a bright GFP variant with optimized cell-filling properties (Campbell et al., 2020), allowed for precise detection of dendritic spines with GFP filling spine necks and heads (Fig. 8C). We developed an image analysis tool to measure local dendrite thickness, count spines and determine the size of spine heads (see Materials and Methods). Fig. 8C shows the segmentation of the dendrite and dendritic spines. Expression of STIM2.2 or STIM2.3 did not significantly change the mean local dendrite diameter (mean ±s.e.m.: 0.78±0.017 μm for control, 0.81±0.05 μm for STIM2.2 and 0.78±0.02 μm for STIM2.3). However, although expression of STIM2.2 slightly increased spine density (Fig. 8D), only expression

of STIM2.3, and not of STIM2.2, significantly increased the mean spine head area, indicating morphogenetic changes (Fig. 8E). These splice-specific effects were seen in three independent transfections: the superplot and statistics shown in Fig. 8E are based on averages of coverslips (triangles) from the different days containing equal numbers of analyzed spines. To avoid misclassification of the shapes of spines (mushroom, stubby or thin), total spine area was analyzed.

## DISCUSSION

The present study confirmed the existence of a predicted short STIM2 variant (NM_001169117.2), here named STIM2.3, as a unique splice variant with splice donor and acceptor sites arising late in evolution and with a short specific sequence that is detected in Hominoidea and in the only living member of the genus *Theropithecus*, namely *Theropithecus gelada*, as well as in one species of Hylobatidae (all classified as Old World primates), as also recently confirmed by Recinos et al. (2024). A beneficial effect of splicing has been shown for a human-specific variant of the Rho GTPase activating protein 11B (ARHGAP11B), which evolved by a single splice site mutation. This variant has been shown to be causative for human neocortical expansion by enhancing basal progenitor cell amplification (Florio et al., 2015, 2016; Xing et al., 2021). Although we found the highest expression of the STIM2 alternative spliced-in exon 13 within the cerebellum, selected other tissues also show expression of STIM2.3; however, quantification from bulk RNA seq data can be misleading due to different relative proportions of cell types, quality of postmortem RNA, short reads, and discrepancies between isoform and junction-level information, potentially due to variant ENST00000477474.3 sharing parts of the genomic region (Fig. S1; www.gtexportal.org). We did detect exon 13 expression in all investigated brain regions, with expression at later developmental stages than *ARHGAP11B*, potentially correlating with the beginning of synapse formation and pruning. Exact staging and cell type specificity is difficult due to the scarcity of developing human brain material. Cerebellar expansion rate significantly increased relative to neocortex in the phylogenetic branch of apes compared to related non-ape branches, indicating a role of cerebellar specialization in cognitive evolution including technical complexities such as production and use of tools and learning of complex motor skills (Barton and Venditti, 2014). While it is unlikely that STIM2.3 has a major role in this expansion, given its low overall frequency, it is possible that selected neuronal subtypes or cells derived from other tissues show much higher splice frequencies. STIM2.3 might be of pathophysiological relevance, as analysis of gene splicing information made available by Elorza et al. derived from postmortem probes of HD patients revealed that usage of the specific splice borders of alternative exon 13 was significantly lower in the striatum of HD patients compared to healthy controls (Elorza et al., 2021) (Fig. 2H). The overall percentage of striatal exon splice inclusion correlates well with our data shown in Fig. 2E. Although steady progression of motor dysfunction is a hallmark of HD, it is not well understood how CAG repeats within the Huntingtin gene lead to motor dysfunction (Smith et al., 2000); however, a compelling correlation between Purkinje cell loss in the neocerebellum and the HD motor symptom phenotype has been found (Singh-Bains et al., 2019).

Functionally, we found that splice-induced deletion of 159 C-terminal residues of STIM2.2 increased, rather than reduced, SOCE. Deletion of only the PBD in full-length STIM2.2 significantly reduced SOCE, confirming previous findings (Ong et al., 2015; Subedi et al., 2018). In STIM1, deletion of only the PBD leads to retention of activated STIM1 at the microtubular network (Chang

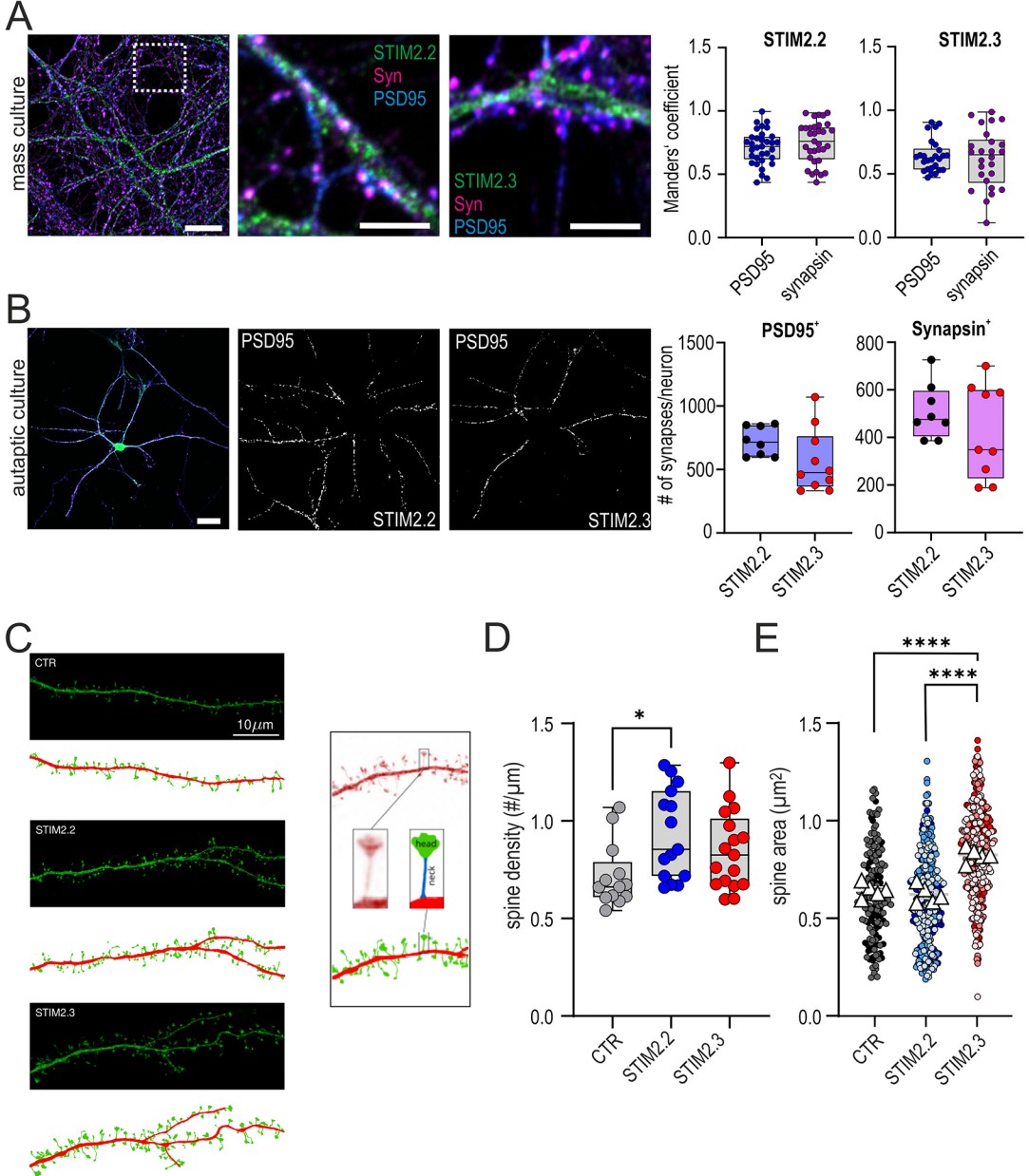

**Fig. 8. STIM2 isoform localization in neurons and isoform-dependent effects on dendritic spines**. (A) Representative images of neuronal mass cultures stained with anti-HA–488 (to label STIM2 isoforms, green), anti-PSD95–568 (displayed in blue color) and anti-synapsin–647 (Syn, displayed in magenta). Dashed box indicates relative size of areas magnified in panels to the right, here shown for STIM2.3. Scale bars: 10 µm. Colocalization analysis of STIM2 isoforms with PSD95 or synapsin. For each condition, 26–33 regions of interests from six coverslips of two independent preparations were analyzed using Manders' overlapping coefficient. STIM2.2 and PSD95, 0.71; STIM2.2 and synapsin, 0.75; STIM2.3 and PSD95, 0.64; STIM2.3 and synapsin, 0.61. (B) Representative image (left) and binary masks (middle, right) of autaptic cultures transduced with STIM2 isoforms and stained with anti-HA–488 (to label STIM2 isoforms), anti-PSD95–568 (blue) and anti-synapsin–647 (magenta). Representative image shows STIM2.3-transduced neurons. Binary masks show PSD95 in neurons transduced with the indicated STIM2 isoform. Scale bar: 20 µm. Quantification of PSD95- or synapsin-positive synapses in STIM2.2- and STIM2.3-transduced neurons. (C) Representative images of dendrites expressing mGreenLantern and the indicated STIM2 variants (top panels) and processed images from the algorithm described in Materials and Methods (lower panels). CTR, mock-transfected culture. Right: illustration of image processing to identify spine head area. (D,E) Quantification of (D) spine density and (E) spine head area from two control transfections and three independent STIM2 isoform transfections, with 106 spines analyzed per condition. In E, each transfection is shown with differentially colored dots, and triangles indicate averages of coverslips, which were used for statistical analysis; total mean±s.e.m. is indicated. *$P<0.05$ (D, Kruskal–Wallis ANOVA with Dunn's multiple comparisons test). ****$P<0.0001$ (E, ordinary one-way ANOVA with Tukey's multiple comparisons test). Data in A,B,D are represented as box and whisker plots. The box marks the 25th to 75th percentiles, with the median indicated by a line. Whiskers show the minimum to maximum range, with all points indicated.

et al., 2018; Liou et al., 2007), suggesting a 'tug of war' between PM attachment and dynamic microtubular retention for STIM1. For STIM2.2 this effect is likely more prominent as STIM2.2 contains two EB-binding sites (EB1/EB3) but also a longer and more dominant PBD (Son et al., 2020). Indeed, simultaneous mutation of

both EB-binding sites (2×IP) and the PBD (Δ5K) in STIM2.2 (STIM2.2 2×IP+Δ5K) rescued the negative effect of STIM2.2Δ5K on SOCE, demonstrating competitive functions of the PBD and EB-binding motifs related to previous findings for STIM1. However, STIM2.3 still showed increased SOCE compared to STIM2.2

2×IP+Δ5K, whereas a deletion of residues 625–746 (STIM2.2Δ624) resulted in SOCE similar to that of wild-type STIM2.2 or STIM2.2 2×IP+Δ5K) (Fig. S3). An increase in STIM1 function upon larger C-terminal deletion has been described for the tubular aggregate myopathy (TAM)-associated gain-of-function mutation STIM1[I484R], which leads to termination of translation at residue 502, also abrogating microtubule-binding sites and the PBD (Kim et al., 2022). Here, interpretation of the activated phenotype (which is only observed with high ORAI1 levels) involves a potentially disabled binding of the inhibitory domain of STIM1 to the CC1 region and disabled binding of STIM1 to SARAF, an accessory protein that maintains STIM proteins in the closed and inactive conformation (Palty et al., 2012; Zomot et al., 2021), or a combination of both. Interaction of SARAF with STIM1 entails a second binding site encompassing residues 490–530 of STIM1 (Jha et al., 2013), a domain that shows only 15% conserved residues between STIM1 and STIM2. Thus, we believe that the mechanism leading to the increased function of STIM2.3 is independent of SARAF, different from the mechanism described for STIM1 and can also be seen in a cell line with endogenous low ORAI1 expression. The region between residues 588 and 623 might be causative for suppression of the enhanced function when comparing STIM2.3 and STIM2.2Δ587 with STIM2Δ624; however, internal deletion of this region from full-length STIM2.2 further decreased function (compare Fig. 4 and Fig. S3). As this region contains eight putative phosphorylation sites, point mutations instead of deletion of amino acids 588–623 (Fig. S3), which potentially causes structural problems, might uncover the nature of the decreased function. An additional difference between STIM1 and STIM2 lies in the $Ca^{2+}$-binding affinity of their respective intraluminal EF hands (Stathopulos et al., 2009), resulting in considerable unfolding and pre-clustering of STIM2 at the PM. This pre-clustering at the PM was reduced but not entirely absent for STIM2.3 and also for all mutants lacking the PBD, confirming previous results with swapped C-terminal domains between STIM1 and STIM2 (Son et al., 2020), and indicates a dominance of the STIM2 PBD over its EF hand regarding pre-clustering at the PM. Additionally, we find that in the absence of endogenous STIM1 but with endogenous ORAI proteins present, all mutants lacking the PBD show reduced cluster sizes after store depletion, confirming that the PBD enhances cluster formation and determines their size and/or stability (Fig. 6), but also clearly indicating that cluster presence and size do not correlate well with overall SOCE (Fig. 7C,E).

Downstream effectors of SOCE include activation and nuclear translocation of the transcription factor NFATc1 (and activation of other transcription factors), which is also of critical importance for neurons. Here, morphological remodeling of axon terminals during synaptogenesis requires calcineurin–NFAT-dependent gene expression (Yoshida and Mishina, 2005). In heterologous expression, STIM2.3 abrogated the constitutive, store depletion-independent nuclear NFATc1 translocation seen with STIM2.2, but enabled similar NFATc1 translocation after store depletion, as NFATc1 translocation saturates at relatively low $Ca^{2+}$ levels. In contrast to deletion of the PBD within STIM1, which abrogates NFAT translocation also after stimulation (Knapp et al., 2022; Ong et al., 2015), we found for STIM2 that as long as intracellular $Ca^{2+}$ reached a critical threshold, NFATc1 translocation was enabled, despite a lack of the PBD, which for STIM2.2Δ5K also required deletion of the EB-binding sites (Fig. 7B). This is in line with results showing that recruitment of AKAP79 (also known as AKAP5) as an NFAT locator requires the N-terminal anchoring domain within ORAI1 (Kar et al., 2021).

AMPK is a master regulator of cellular energy metabolism as it senses and monitors the concentration of the nucleosides AMP, ADP and ATP, which all bind competitively to the γ subunit of the heterotrimeric αβγ complex of the active enzyme. Previous work has shown that STIM2 interacts with AMPK (Chauhan et al., 2019; Stein et al., 2019) and is also a substrate for AMPK regulation, leading to reduced SOCE activity by phosphorylation (Nelson et al., 2019; Stein et al., 2019; reviewed in Poth et al., 2020). Our results (Fig. 7F–H) show a decreased interaction of STIM2.3 both with AMPK and with active phospho-AMPK. Moreover, one of the major AMPK substrate residues within STIM2, namely Ser680, is lacking in STIM2.3, potentially alleviating SOCE inhibition by AMPK-dependent phosphorylation, which might contribute to the increased SOCE phenotype of STIM2.3, although STIM2.2Δ624 shows no increased SOCE (Fig. S3). In neurons, hyperactive AMPK can cause metabolic stress and impair neuronal polarization (Ramamurthy et al., 2014; Williams et al., 2011), which might be reduced by STIM2.3 due to reduced AMPK interaction. Future work is necessary to delineate the crosstalk between STIM2.2 and STIM2.3 and AMPK activity in neurons.

In contrast to a partially increased presynaptic localization of the short STIM1B splice variant (Ramesh et al., 2021), we found neither differential localization of STIM2.3 when expressed in neuronal cultures, nor differences in dendrite length or in the number of synapses counted from autaptic cultures. Overexpression of STIM2.2 leads to an expected increase in spine density (Fig. 8D), as reported previously (Pchitskaya et al., 2017; Popugaeva et al., 2015). However, a significant increase in the area of spine heads is found only upon overexpression of STIM2.3, and not of STIM2.2, in hippocampal neurons. This finding is initially surprising given the reports by Pchitskaya et al. (2017) showing that EB3 (also known as MAPRE3) and STIM2.2 are necessary for normal dendritic spine morphology. However, the same authors found that STIM2.2 overexpression does not change the fraction of mushroom spines. A recent report by Rakovskaya et al., which shows that STIM2 constructs with deleted EB-binding sites lead to decreased SOCE (not seen in our hands using SHSY-5Y cells, Fig. 4) and a reduction in the number of spines containing the spine apparatus with ER cisternae (Rakovskaya et al., 2025), initially appears contradictory to our findings, as STIM2.3 also lacks EB-binding sites. However, the STIM2.3 gain-of-function phenotype with increased SOCE (Figs 3,4), which is also evident upon co-expression with STIM2.2 (Fig. S2D,E), allows for a co-transport of STIM2.3–STIM2.2 dimers along dynamic microtubules and might explain how $Ca^{2+}$ influx, gene expression and dynamic microtubule attachment shape STIM2-dependent spine morphology. In addition, the smaller size of STIM2.3 might allow for a facilitated diffusion into spines containing smooth ER, which has been shown to be critical for NMDA-dependent chemical long-term potentiation (cLTP; Borczyk et al., 2019). Physiologically, regulated STIM2 splicing in primates might thus allow for an additional boost of SOCE and potentially splice-specific gene expression to increase the fraction of large spine heads fortifying synaptic efficiency. In the context of the recently reported regular spaced dendritic ER–PM junctions containing stable STIM2.2 (Benedetti et al., 2025), the shorter STIM2 variant might enable microtubule-independent diffusional mobility within the ER as well as boost local $Ca^{2+}$ influx.

In summary, our work identifies STIM2 splicing as a new regulator of SOCE in the brain. Its evolutionarily late and brain-specific addition to the STIM repertoire, with increased SOCE and potentially store (activity)-dependent gene regulation, leads us to hypothesize that STIM2.3 presents an additional regulator within

specified neurons that, by increasing dendritic spine morphology, might strengthen synaptic facilitation.

## MATERIALS AND METHODS
### Cell culture and transfection
HEK cells (ATCC, CRL-1573) and SH-SY5Y cells (ATCC, CRL-2266) were cultured at 37°C, 5% $CO_2$ in a humidified incubator in their respective medium [Dulbecco's modified Eagle's medium (DMEM) for HEK STIM1/2$^{-/-}$ (STIM1/2 DKO; Zhou et al., 2018); DMEM with 1% non-essential amino acids (NEAA) for SH-SY5Y STIM2$^{-/-}$ and STIM1/2$^{-/-}$ (Gibco)]. All cell culture media were supplemented with 10% fetal calf serum (FCS; Gibco). Cells were passaged weekly and detached using trypsin. For imaging experiments, HEK STIM1/2 DKO cells were transfected via electroporation using the Amaxa Nucleofector II (Lonza) according to the manufacturer's protocol. For co-immunoprecipitation, HEK STIM1/2$^{-/-}$ cells were transfected using JetOptimus (Polyplus) according to the manufacturer's protocol. SH-SY5Y cells were transfected using JetPrime (Polyplus transfections) transfection reagent according to the manufacturer's protocol. Cells were analyzed 24 h post transfection. Table S1 lists all utilized constructs and vector backbones.

### Generation of stable cell line
To generate a cell line stably expressing HA–STIM2.3, SH-SY5Y STIM2$^{-/-}$ cells were transfected with plasmid encoding HA–STIM2.3 and a G418 resistance gene. Cells were seeded into 60 mm dishes containing DMEM with 10% FCS and 1% NEAA 24 h prior to transfection using JetPrime according to the manufacturer's protocol. 24 h post transfection, medium was changed to DMEM with 10% FCS, 1% NEAA and 1 mg/ml G418 as the selection antibiotic. The selection medium was changed twice weekly until constant growing was observed. Stable overexpression of HA–STIM2.2 or HA–STIM2.3 was analyzed using $Ca^{2+}$ imaging and western blotting. SH-SY5Y STIM2$^{-/-}$ cells were generated as described for double KO (STIM1$^{-/-}$; STIM2$^{-/-}$) in Ramesh et al. (2021).

### PCR and qRTPCR
RNA isolation was performed using TRIzol (Life Technologies) according to the manufacturer's protocol. For cDNA synthesis, SuperScriptII Reverse Transcriptase (Life Technologies) was used. qRT-PCRs were performed using QuantiTect SYBR Green Kit (Qiagen) and a CFX96 Real-Time System (Bio-Rad). Expression of genes of interest were analyzed by normalization to TBP and RNAPol (not shown) using the $\Delta C_q$ method. Values are shown as $2^{-(\Delta C_q)}$. Table S2 lists the applied primers.

### Co-immunoprecipitation, western blot analysis and antibodies
For harvesting cells for protein analysis, transfected cells were washed with ice-cold PBS, detached in ice-cold RIPA buffer containing 150 mM NaCl, 50 mM Tris-HCl, 1% Nonidet P40, 1% Triton X-100 (Merck) and Protease Inhibitor Complete (Sigma-Aldrich) using a cell scraper. Cells were lysed by freeze (for 5 min at −80°C) thawing, then vortexed and centrifuged at 21,380 $g$ (4°C) for 20 min. For co-immunoprecipitation, transfected cells were washed in ice-cold PBS, detached in immunoprecipitation buffer containing 20 mM Tris-HCl, 100 mM KCl, 10% glycerol (v/v), 0.5% n-dodecyl β-D-maltoside (DDM) using a cell scraper. Cells were lysed for 30 min at 4°C with end-over-end tumbling and then centrifuged at 18,000 $g$ (4°C) for 30 min. Protein concentrations were determined using BCA reagent (Thermo Fisher Scientific) according to the manufacturer's protocol. Co-immunoprecipitation was performed by incubating 1 mg cell lysate protein with 30 µl Pierce anti-HA agarose overnight with end-over-end tumbling at 4°C using protein low-binding tubes. Protein-bound agarose was washed four times for 5 min in ice-cold lysis buffer and centrifuged for 30 s at 500 $g$ before being eluted with 50 mM Tris-HCl pH 6.8 and 2% SDS for 5 min at 90°C. Protein fractions were denatured in Laemmli buffer at 65°C for 15 min. Discontinuous SDS-PAGE was performed with subsequent electrotransfer to nitrocellulose or PVDF membrane for 90 min at 150 mV and 4°C. Membranes were blocked in 5% skimmed milk in TBS-T buffer (50 mM Tris-HCl pH 7.4, 0.15 M NaCl, 1% Tween 20) for 60 min. Primary antibodies (α-HA, 3F10, Roche) were incubated overnight at 4°C in PBS containing 1% bovine serum albumin

(BSA) and 0.02% sodium azide. Membranes were washed three times for 10 min in TBS-T and incubated with the secondary antibody in blocking buffer for 60 min at room temperature. After washing three times for 10 min, the membranes were incubated in Clarity Western ECL Substrate (Bio-Rad) and detection of chemiluminescence was performed on a CCD camera equipped ChemiDoc XRC+ system (Bio-Rad). Densitometric quantification of non-saturated signals was done using ImageLab software (Bio-Rad). Primary and secondary antibodies are listed in Table S3. Additional blots underlying quantification of bound AMPK are shown in Fig. S6 (blot transparency file). The blots shown are not necessarily those exposure times used for quantification, as we only used non-saturated signals for quantification.

### Fluorescence-based $Ca^{2+}$ imaging
HEK STIM1/2$^{-/-}$ cells were seeded onto 25 mm coverslips in 35 mm dishes containing cell culture medium after transfection with 1 µg YFP-, HA- or mKate2-tagged STIM2.2 or STIM2.3 (all pEX) using electroporation. SH-SY5Y STIM2$^{-/-}$ or STIM1/2$^{-/-}$ cells were seeded in 35 mm dishes 24 h prior to transfection with 2 µg of YFP-, HA- or mKate2-tagged STIM2.2 or STIM2.3 (all pEX). Transfected cells were reseeded 5 h post transfection onto 25 mm coverslips. After 16–24 h, cells were loaded with 1 µM Fura-2 in DMEM and 10% FCS (SH-SY5Y: additionally with 1% NEAA) at room temperature for 30 min, transferred to a perfusion chamber and washed with external $Ca^{2+}$ Ringer solution containing 155 mM NaCl, 2 mM $MgCl_2$, 10 mM glucose, 5 mM HEPES and either 0.5–1.5 mM $CaCl_2$ or no $CaCl_2$, but 1 mM EGTA and 3 mM $MgCl_2$ instead. ER stores were depleted using 1 µM thapsigargin (TG) in 0 mM $Ca^{2+}$ ($Ca^{2+}$ re-addition) or 0.5–1.5 mM $Ca^{2+}$ (global) Ringer solution. Image acquisition was performed with a Zeiss Axio Observer A1 inverted microscope, Polychrome V (Till Photonics) light source and Clara CCD camera (Andor). F76-521 Fura-2 HC, F36-528 YFP and F20-451 RFP filter cubes (AHF Analyzetechnik AG) were used. Fluorescence emission at 510 nm with excitation at 340 nm and 380 nm was acquired with 10–30 ms exposure times and 1×1 or 2×2 pixel binning through a 20×0.75 Fluar (Zeiss) objective. Acquisition was monitored by, and background-subtracted 340 nm/380 nm ratio values were calculated using, VisiView software (Visitron Systems). Fluorescence ratio data was further analyzed using IgorPro (Wavemetrics). For kinetic parameters, average (5–10 frames) basal and plateau ratio values and maximal TG- or $Ca^{2+}$ re-addition-induced peak ratio values were determined. To obtain Δ ratios, the average ratio before addition of TG or $Ca^{2+}$ was subtracted, respectively. The influx rate is represented by the slope of a linear fit performed on the ratio values of 1 frame before and 4 frames after $Ca^{2+}$ re-addition. For quantification of global $Ca^{2+}$ entry, the area under the curve (AUC) upon store depletion was determined. The data were then imported into GraphPad 8.4 (Prism) for further statistical analysis and plotting.

### Live-cell imaging
HEK STIM1/2 DKO cells were transfected with STIM2.2–mCherry and either YFP–STIM2.3, YFP–STIM2.2Δ5K or YFP–STIM2.2Δ587. Fluorescence was detected using the wide-field epifluorescence microscope cell observer A1 (Zeiss) with the Fluor 100×/ oil objective. To detect YFP, fluorescence filter cube 54HE and LED 470 Ex (470/40) were used, and for mCherry detection, filter cube 56HE and LED N-White+Ex (556/20) were used. Images were taken as Z-stacks and processed as a maximum intensity projection (MIP) for visualization. Colocalization was analyzed using Fiji (https://fiji.sc/) plugin JACoP from a single stack. Threshold was set as mean plus two times the s.d.

### TIRF microscopy for cluster analysis
Cluster formation was monitored using a Leica TIRF MC system. HEK STIM1/2 DKO cells were seeded into 35 mm dishes 24 h prior to transfection with PH-PLCγ–mCherry (Nicolas Touret, University of Alberta, Edmonton, Canada) and YFP–STIM2.2, YFP–STIM2.3, YFP–STIM2.2Δ5K or YFP–STIM2.2 2×IP+Δ5K. After 4 h of incubation, cells were reseeded onto 25 mm coverslips in 35 mm dishes containing DMEM and 10% FCS. 24 h after transfection, coverslips were transferred to a perfusion chamber and washed with 0.5 mM $Ca^{2+}$ Ringer solution. Store depletion and subsequent cluster formation was induced after 60 s baseline acquisition by perfusion with 0.5 mM $Ca^{2+}$ Ringer solution containing

1 µM TG. Images were taken with a 100×1.47 oil HCX PlanApo objective. YFP and mCherry were excited using 488 nm or 561 nm lasers with BP525/50 and DRT emission filters, respectively. The TIRF focal plane was set according to the PH-PLCγ–mCherry signal, and a penetration depth of 90 nm was chosen.

## NFAT-c1 assay
SH-SY5Y STIM1/2$^{-/-}$ cells were seeded in 35 mm dishes 24 h prior to transfection with GFP-tagged NFATc1 (NFAT1–GFP) and mKate2–STIM2.2, mKate2–STIM2.3, mKate2–STIM2.2Δ5K or mKate2–STIM2.2 2×IP+Δ5K. After 5 h of incubation, cells were reseeded onto 25 mm coverslips in 35 mm dishes. 24 h post transfection, NFAT1–GFP signal in the cytoplasm versus nucleus was detected on the Zeiss Axio Observer A7 equipped with a Prime95B sCMOS camera (Photometrics) using a 63×/1.40 Plan Apochromat oil objective. YFP and mKate2 signals were excited using an HXP 120 V compact light source with 480-498 nm/560-584 nm excitation filters and Fura8 (BS488 nM, BP500-550 nm)/63 HE Red (BP559-585 nm, BS590 nm, BP600-690 nm) emission filter cubes. Cells were imaged in a Z-plane near the nucleus stained with 1 mg/ml Hoechst 33342 dye immediately before the measurement. Cells were activated using 1 µM TG in 1 mM Ca$^{2+}$ Ringer solution. The ratio of GFP signal in the cytoplasm versus in the nucleus was determined using Fiji.

## BiFC
For the BiFC assay, a FACSVerse system (BD Biosciences) was used to detect YFP fluorescence in viable cells. To this end, HEK STIM1/2 DKO cells (Zhou et al., 2018) were transfected in a 6-well scale with HA–STIM2.2 or HA–STIM2.3 fused to amino acids 156–720 of YFP (YFPc) in pEX as bait in combination with the protein of interest fused to amino acids 1–155 of YFP (YFPn) in pBabe or pMax as prey. Transfection was performed at equimolar ratios of DNA using a total amount of 2 µg DNA. To induce SOCE, 1 µM TG was added to the medium and incubated for 10 min 24 h post transfection. After detaching cells in PBS containing 1 µM TG, cells were centrifuged at 320 $g$ in FACS tubes at 4°C for 10 min and resuspended in 400 µl ice-cold FACS buffer (5% FCS, 0.5% BSA and 0.07% NaN$_3$ in PBS) with additional Zombie Aqua (1:1000 Fixable Viability Kit, BioLegend) to stain for vital cells. BD FACSuite and FloJo 10.0.7 (BD Biosciences) were used for determination of the percentages of YFP-positive cells.

## Bioinformatics
The following three datasets were downloaded from the NCBI Sequence Read Archives (SRA): (1) SRP331938, GSE181813, representing six male and six female non-alcohol use disorder postmortem control donors, age not listed, RNA integrity number (RIN) values 6–9; (2) SRP346150, analyzing GSE207713, with iPSC-derived astrocytes from four control donors; and (3) GSE188847, which analyzes aged COVID-19-unaffected postmortem controls (including six males and four females), age range 45–64 years. Additional datasets were obtained for development stages Carnegie stage 22 and 9 post-conception weeks, downloaded from the Biostudies database, a data infrastructure belonging to The European Bioinformatics Institute https://www.ebi.ac.uk/biostudies/arrayexpress/studies/E-MTAB-4840 (Lindsay et al., 2016). HD and control

A bioinformatics workflow was constructed to first quantify gene expression and, secondly, to analyze splicing events for selected genes. Files in FASTQ format for all datasets were downloaded and used as input for the nf-core *rnaseq* pipeline to map and quantify reads abundance using Salmon's fast mapping algorithm (Patro et al., 2017). Mapped reads in BAM format were sequentially subjected to splicing variations detection by MAJIQ software. Among the current tools for splice events study, MAJIQ is a comparably fast and reliable tool (Mehmood et al., 2020). MAJIQ yields a Percent Selected Index (PSI) for each recognized and *de-novo* splice junction as quantification measurements for splice variations, which efficiently facilitates our current study to identify and compare STIM2 alternative events across various samples. For both mapping and splicing analysis stages, the Human Reference Genome build 38 (hg38) retrieved from NCBI in GTF format was used. For easier comparison to qRTPCR data, single exon 13 splice border probabilities were averaged to give the overall exon 13 inclusion probability. Further RNA seq data (Fig. S1)

was obtained from ENCODE (see Table S4 for accession codes). Isoform-level expression was performed using the tools Salmon (https://github.com/COMBINE-lab/salmon) and Kallisto (https://github.com/pachterlab/kallisto); junction-level quantification used the tool rMATS (rnaseq-mats.sourceforge.net). RNA-Seq datasets underlying data in Fig. S2C were deposited into the ArrayExpress collection at EMBL-EBI under the accession number E-MTAB-12747.

## Isolation of postmortem brain tissue
In accordance with the ethics statement below, tissues from body donors were extracted by the anatomist (T.T.) 8–12 h postmortem. 50–100 mg tissues were flash-frozen in liquid N$_2$ and homogenized into powder to which 800 µl–1 ml TRIzol was added and RNA was isolated according to the manufacturer's protocol. 260 nm/280 nm absorbance ratios and/or RIN values were used for quality control. No good correlation between RIN value and presence/absence of STIM2.3 was observed. See Table S4 for additional description and references pertaining to Fig. S1.

## Ethics statement
Ethical permission and written informed consent was previously obtained to receive postmortem tissues and granted by the ethics commission of the medical board of the Saarland, Nrs. 163/20 and 148/18.

## Animals
All experimental procedures involving the use of animals were approved and performed in accordance with EU guidelines, licences and the ethic regulations of the animal welfare committee at the University of Saarland.

## Primary neuronal cultures
Hippocampal neurons and cortical astrocytes were prepared from C57BL/6N mice of both sexes at postnatal day 0–1 (P0–P1), and protocols were loosely based on Jones et al. (2012), Burgalossi et al. (2012), and Wolfes and Dean (2018). Autaptic neuronal cultures were cultured as previously described (Schwarz et al., 2017). For mass cultures, neonatal pups were decapitated, and the brain transferred to ice-cold Hibernate-A medium (Gibco, A1247501). Meninges were removed, and the hippocampi (for neuronal cultures) or cortices (for astrocyte cultures) were dissected. Hippocampi were pooled and transferred to a digestion solution containing 20 U/ml Papain (Worthington Biochem, LS003124), 1 mM CaCl$_2$ (Sigma), 1.65 mM L-cysteine (Sigma), 0.5 mM EDTA (Sigma) and 0.1 mg/ml DNAse I (Calbiochem, 260913) in DMEM (Gibco). After 20 min incubation at 37°C the solution was replaced with a stop solution containing 2.5 mg/ml BSA (Pan Biotech), 2.5 mg/ml Trypsin-Inhibitor (Sigma T9253), 10% FCS (Gibco) and 0.1 mg/ml DNAse I in DMEM for 2 min at 37°C. The tissue was washed once with DMEM containing 10% FCS and 1× penicillin streptomycin (P/S; Gibco) then mechanically triturated using P1000 and P200 pipette tips. The cell suspension was passed through a 40 µm cell strainer (Greiner EASYSTRAINER) and cells plated onto 0.1 mg/ml poly-D-lysine (PDL; Sigma, P6407)-coated 18 mm coverslips inside 12-well plates. After 1–2 h, coverslips were transferred with cells facing upwards to previously prepared 12-well plates containing a 60–80% confluent cortical astrocyte feeder layer in Neurobasal A (Gibco) with 2% B27 (Gibco), 1% GlutaMAX (Gibco) and 1% P/S, using paraffin dots as spacers. On day *in vitro* (DIV) 2, cytosine β-D-arabinofuranoside (Sigma, C1768) was added to a final concentration of 2 µM with a partial medium exchange. On DIV7, 2 µl of crude AAV2/1-containing supernatant encoding mKate2- or HA-tagged STIM2.2 or STIM2.3 driven by the CaMKIIα promotor was added with a partial medium exchange. Experiments were conducted at DIV14–21. Astrocyte cultures were prepared by mincing the obtained cortices and incubating them in 0.05% Trypsin-EDTA (Gibco) solution for 20 min at 37°C. Cells were washed five times with ice-cold PBS (Gibco), once with DMEM containing 10% FCS and P/S, and triturated using P1000 and P200 pipette tips. The cell suspension was passed through a 100 µm cell strainer (Greiner) and centrifuged for 10 min at 250 $g$. The cell pellet was resuspended in DMEM containing 10% FCS and 1% P/S and plated in T-75 flasks coated with 0.05 mg/ml PDL. The medium was exchanged every other day, and once reaching confluency, the flask was shaken for 2–6 h at 180 rpm inside an incubator and washed with PBS to remove non-astrocyte cells such as microglia or neurons. The astrocytes were

detached using 0.25% Trypsin-EDTA, resuspended in DMEM containing 10% FCS and P/S then counted. After centrifugation for 10 min at 250 *g*, cells were resuspended in FCS containing 10% DMSO at $2 \times 10^6$ cells/ml and cryo-conserved by slow cooling overnight and subsequent storing in the vapor phase of liquid nitrogen for up to 6 months. An aliquot of cryo-conserved astrocytes was quickly thawed in a 37°C water bath, the cell suspension was dropped into DMEM containing 10% FCS and P/S medium, and then centrifuged for 10 min at 250 *g*. The pellet was resuspended in DMEM containing 10%FCS and P/S, and ~80,000 cells per well were plated in 12-well plates containing paraffin dots and coated with 0.05 mg/ml PDL. The medium was changed daily, and a day before the expected date of neuron preparation exchanged to Neurobasal A containing 2% B27, 1% GlutaMAX, 1% P/S for conditioning. For mass culture immunofluorescence experiments investigating dendritic spine morphology, hippocampal neuron cell suspension was prepared as described above and ~80,000 viable cells in 50 µl plating medium were plated directly onto PDL-coated 18 mm coverslips, and 1 ml of Neurobasal A medium containing 2% B27-Plus, 1% P/S and 1% GlutaMAX was added after 30 min. Hippocampal cultures were sparsely transfected using the ProFection Mammalian Transfection System (Promega) with either 1 µg pAAV-CAG-mGreenLantern alone or in combination with 1 µg pAAV-CaMKII-mKate2-Stim2.2 or Stim2.3 at DIV5–7 and used for experiments at DIV15–21.

## Small-scale AAV supernatant production
Small-scale production of AAV-containing medium was based on Vandenberghe et al. (2010). HEK293T cells (CVCL_0063) were plated in DMEM containing 10%FCS and P/S in 6-well plates with a density of 800,000 cells/well. After 24 h, cells were triple transfected with 1.5 µg pADeltaF6 (Addgene #112867), 0.75 µg pAAV 2/1 (Addgene #112862) and 0.75 µg expression plasmids using jetPEI (PolyPlus Transfection) using a 1:2 DNA:jetPEI ratio. After 24 h, the medium was exchanged with 1 ml Neurobasal A containing 2% B27, 1% GlutaMAX and 1% P/S. After 72 h cells were scraped off, centrifuged for 10 min at 250 *g* and the supernatant saved. The cell pellet was subjected to three freeze–thaw cycles in liquid nitrogen, resuspended in 500 µl medium, cleared by centrifugation at 1000 *g* for 10 min and pooled with the previous supernatant. The AAV-containing supernatant was stored at −80°C, and required amounts for transductions were determined empirically.

## Immunohistochemistry of neuronal cultures
Mass and autaptic cultures were fixed in PEM buffer (80 mM PIPES pH 6.8, 5 mM EGTA and 2 mM MgCl₂) containing 4% paraformaldehyde (PFA) or in 4% PFA and 4% sucrose in PBS for 10–15 min at room temperature. Unspecific epitopes were blocked using 1×PBS containing 0.1% Triton X-100 and 0.22% cold water fish skin gelatin (Sigma, G7041), or 10% normal goat serum (NGS; Jackson ImmunoResearch) with 0.1% Triton X-100 in PBS for 1 h at room temperature. Primary and secondary antibodies (Table S3) were diluted in blocking solution or 5% NGS and 0.1% Triton-X100 in PBS and incubated overnight at 4°C (primary antibody) and for 90 min at room temperature (secondary antibody). Cells were mounted using Prolong Glass Mounting medium (Thermo Fisher Scientific, P36984). Images were acquired as a Z-stack on a LSM 800 with a ×63/1.4 NA oil objective (mass cultures) or a ×40/1.4 NA oil objective (autaptic cultures). Images were analyzed as MIPs. The background was removed using rolling-ball background subtraction with the radius set to the approximate diameter of the largest cluster. Colocalization was analyzed using Fiji plugin JACoP from a single stack. Threshold was set as mean plus two times the s.d. Before the number of PSD95- or synapsin-positive synapses was determined, the Fiji watershed algorithm was applied to binary images. The 'Analyze Particles' tool of Fiji with the size and circularity set to 0.01–5.00 µm and 0.1–1.00, respectively, was used to determine the area of synapses. Main dendrite length was analyzed using the plugin NeuronJ (Meijering et al., 2004). The data were imported into GraphPad 8.4 (Prism) for further statistical analysis and plotting.

## Dendrite image processing and data analysis
Grayscale images were converted into two-dimensional data arrays using MATLAB. Background noise was subtracted in multiple steps: an intensity threshold was applied to filter weak noise, the remaining signal was converted to a binary format, and the Hoshen–Kopelman (HK) cluster detection algorithm (Hoshen and Kopelman, 1976) was used to remove small disconnected islands. The remaining large islands, representing dendrites and their connected spines, were processed to identify spines with high accuracy. The primary spine identification algorithm iteratively classified and removed border pixels from the binary matrix, progressively thinning structure by peeling away boundary elements, including protrusions (spines). This thinning process eventually severed spine necks from the dendritic channel. After eliminating all spine necks and applying the HK algorithm to remove isolated spine heads, the procedure was reversed by reconstructing the channel structure. Due to the irreversibility of the process, only the channel was restored, and by subtracting it from the original data array, the isolated spines (including their necks) were obtained. To validate this approach, results were compared against an alternative method in which grayscale dendrite structures were reconstructed following background removal. This method estimated occupied regions around each pixel to identify central pixels and reconstruct the dendritic centerline. Layers were iteratively added to restore the original channel width, and subtracting this reconstructed channel from the original data isolated the spines. Comparing the results of both approaches ensured accurate spine identification. A similar approach was applied to distinguish between spine heads and necks. Finally, each spine area was quantified based on the size of its corresponding island. The mean position of the initial spine neck pixels was used as the reference point, and the local dendritic channel thickness at this location was estimated as the shortest distance required to bisect the channel. To measure the local density of spines, the positions of the reference points for spine necks and the extracted centerline information from the second method were utilized.

## Statistical analysis
Statistical analysis was performed using the software GraphPad Prism 9. The statistical tests were chosen according to data distribution and number of conditions. Significances are as follows: *$P<0.05$, **$P<0.01$, ***$P<0.001$. To improve readability of figures, all statistical data showing $P<0.0001$ (****) are also marked with ***. Data are displayed as box and whisker plots, with the median shown as a line within the box (25th to 75th percentiles), whiskers showing minimum and maximum values, and all data points obtained from three independent transfections.

## Acknowledgements
We thank Drs Marcus Grimm and Tobias Hartmann for initial transfection of SH-SY5Y cells with the STIM2 targeting construct; Drs Beate Winner, Tania Rizo and Nicole Ludwig for additional samples of cDNA and RNA; and Dr Stephan Maxeiner for pointing out additional genomic sequences. We also acknowledge all members of the Niemeyer, Hoth, Bruns and Engel labs for discussions, the use of equipment as well as for further technical help in cell culture.

## Competing interests
The authors declare no competing or financial interests.

## Author contributions
Conceptualization: B.A.N.; Data curation: V.P., H.T.T.D., K.-L.L., K.F., V.H.; Formal analysis: V.P., H.B.R., D.A., V.H., R.S., B.A.N.; Funding acquisition: B.A.N.; Investigation: V.P., B.A.N.; Methodology: V.P., L.J., K.F., D.A., T.T.; Project administration: B.A.N.; Resources: L.J., D.A., T.T.; Software: R.S., V.H.; Supervision: D.A., V.H., B.A.N.; Validation: H.B.R.; Visualization: V.P., B.A.N., L.J.; Writing – original draft: V.P., V.H., B.A.N.; Writing – review & editing: B.A.N.

## Funding
Funding was provided by the Deutsche Forschungsgemeinschaft (D.F.G.) grants SFB894 (157660137, A2), SFB1027 (200049484, C4), TRR219 (322900939, M04) to B.A.N.; SFB1027 (200049484, C3) to V.H.; and SFB1027 (200049484, A7) to R.S. Open Access funding provided by University of Saarland. Deposited in PMC for immediate release.

## Data and resource availability
RNA-Seq datasets underlying data in Fig. S2C were deposited into the ArrayExpress collection at EMBL-EBI under the accession number E-MTAB-12747. All other relevant data and details of resources can be found within the article and its supplementary information.

**Peer review history**
The peer review history is available online at https://journals.biologists.com/jcs/lookup/doi/10.1242/jcs.264353.reviewer-comments.pdf

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
