## [Peer Review File · Journal of Cell Science]

Alternatively spliced STIM2.3 is an evolutionary late store-operated Ca²⁺ entry regulator expressed in brain

Vanessa Poth, Hoang Thu Trang Do, Lukas Jarzembowski, Katrin-Lisa Laius, Kathrin Foerderer, Thomas Tschernig, Hanah Betsy Robertson, Dalia Alansary, Reza Shaebani, Volkhard Helms and Barbara A. Niemeyer

DOI: 10.1242/jcs.264353

Editor: Jennifer Lippincott-Schwartz

Review timeline

Original submission:	27 August 2025
Editorial decision:	13 November 2025
First revision received:	21 January 2026
Accepted:	23 February 2026

Original submission

First decision letter

MS ID#: jcs.264353

MS TITLE: Alternatively-spliced STIM2.3 is an evolutionary late gain-of-function regulator expressed in brain

AUTHORS: Vanessa Poth; Hoang Thu Trang Do; Lukas Jarzembowski; Katrin-Lisa Laius; Kathrin Foerderer; Thomas Tschernig; Dalia Alansary; Reza Shaebani; Volkhard Helms; Barbara A Niemeyer

ARTICLE TYPE: Research Article

Dear Dr Niemeyer,

We have now reached a decision on the above manuscript.

As you will see, the reviewers raise a number of substantial criticisms that prevent me from accepting the paper at this stage. They suggest, however, that a revised version might prove acceptable, if you can address their concerns. If you think that you can deal satisfactorily with the criticisms on revision, I would be pleased to see a revised manuscript. We would then return it to the reviewers.

Reviewer 1

Advance summary and potential significance to field

The manuscript "Alternatively-spliced STIM2.3 is an evolutionarily late gain-of-function regulator expressed in brain" by Poth et al details the discovery and molecular characterization of a novel alternatively-spliced isoform of the store-operated calcium entry (SOCE) channel activator STIM2. The authors demonstrate that the newly discovered splice variant, STIM2.3, is expressed in brain of late-evolving primates, and further show that STIM2.3 lacks 159 C-terminal amino acids but contains 12 unique amino acids compared to wildtype STIM2 (STIM2.2). Notably, the truncation of STIM2.3 includes the two EB1-binding domains and the poly-basic domain of wildtype STIM2.

Detailed structure-function analysis demonstrates that STIM2.3 is a gain-of-function variant of STIM2, though the exact mechanism of gain-of-function could not be delineated. The authors then show that STIM2.3 may play a role in increasing the size of neuronal dendritic spines with potential implications for synaptic strength and efficiency. These findings are significant because they add to our understanding of the molecular and functional complexity of the ubiquitous SOCE signaling mechanism, demonstrating that SOCE may be fine-tuned through alternative splicing in a cell and tissue-specific manner. These findings also shed new light on the role of SOCE in neurons and nervous system function overall, an area that has long been enigmatic in the calcium signaling field.

Comments for the author

Overall, the experiments in this study are well-designed and executed, and the results support the conclusions as described. Most of my suggestions relate to enhancing the clarity of the writing and figures/data presentation and should not require new experiments (with the possible exception of Comment 1 below). The following are my specific suggestions and comments to the authors:

1. The legend for Figure 8 is missing, and the legend labeled "Figure 8" actually pertains to Figure 9. The lack of a legend also made it difficult to review and interpret the data presented in Figure 8. For example, in the pull-down experiments presented in 8A and B, it appears that pEYPC1 is used as an empty-vector control; however, the subsequent IPs utilize HA-tagged proteins and anti-HA as the pull-down antibody. A vector expressing HA alone would be a better empty-vector control instead of EYFP. Also, panel C for -thapsigargin quantifies total AMPKa whereas panel E for +thapsigargin (should this actually be panel D?) quantifies p-AMPKa. If there is a reason for quantifying total AMPKa versus p-AMPKa, it should be better explained. Lastly, the data in panels C and E would be easier to compare and interpret if the y-axes for both graphs were scaled the same.
2. In the graphs in Figures 3B,D and 4C,E, it is not immediately clear how the colors of the bars correspond to specific experimental conditions - it took me some time to figure out that the colors match those in the calcium traces in the preceding panels. I would suggest showing the same color key near the bargraphs as in the calcium traces, or label the bargraphs on the x-axes. In addition, it was not immediately clear that the title labels correspond to the graphs above (e.g., "resting" and "influx rate" in 3B). I would suggest moving the title to the tops of each graph.
3. The microscopy images shown in Figures 5 and 6 would be easier to view and interpret if each single-channel image were shown in grayscale.
4. Similar to above, nuclear vs cytoplasmic NFAT localization is difficult to determine in Figure 7A as shown. I would suggest separating the NFAT and STIM2 channels into component images shown in grayscale. And I would again suggest separating the images in Figure 9 into grayscale, single-channel images.
5. Do the quantifications in Figure 6B and C represent the post-thapsigargin condition? This should be clearly indicated. Also, it is difficult to draw many conclusions from Figure 6C as shown other than that there are fewer 0.05 μm^2 clusters for STIM2.2 compared to the other variants. It seems that there may also be differences for the large cluster sizes (0.6 - 1 μm^2) but the y-axis scale makes it impossible to clearly see this.
6. Several major concepts discussed throughout the paper are never thoroughly introduced, such that readers unfamiliar with the SOCE field may not understand their significance. Specifically, nuclear NFAT localization is analyzed as a readout of downstream SOCE function, but the role of SOCE in driving NFAT nuclear translocation is never explained. I would suggest including the role of SOCE in NFAT regulation in the Introduction. Also, many of the experiments test the role of the SxIP (EB1-binding) and poly-basic (PB) domains of STIM2, but the function of these domains is never fully explained.
7. In line 75 (Results), the authors should clearly define that STIM2.2 corresponds to wildtype STIM2, and utilize the STIM2.2 notation throughout the paper.
8. The sentence beginning on line 275, "Importantly, and unlike usually quantified,..." is unclear and should be reworded for clarity.
9. It would be helpful if the authors would better describe autaptic cultures (i.e., how are they generated, what kind of cells do they include?).
10. Can the authors show representative images in Figure 9 of spines for control, STIM2.2, and STIM2.3 samples?

Reviewer 2*Advance summary and potential significance to field*

In this manuscript by Poth et al., the authors characterized an alternatively spliced variant of STIM2, named STIM2.3. In this STIM2.3 variant, the last 159 amino acids of wildtype STIM2 (STIM2.2) was replaced by 12 amino acids derived from Exon 13. As a result, several regulatory regions in STIM2, including the C-terminal polybasic motif, two microtubule end-binding SXIP motifs, and more than 10 potential serine phosphorylation sites, are missing in STIM2.3. The authors proposed that STIM2.3 is a "gain-of-function" variant based on a stronger thapsigargin-induced calcium response in STIM2.3-expressing cells compared with that in wildtype STIM2-expressing cells. Nevertheless, unlike wild-type STIM2, STIM2.3 was not able to promote NFAT nuclear localization in resting cells, a process dependent on STIM-mediated calcium entry. The authors further generated several STIM2 deletion mutants to test the role of missing regions in STIM2.3. Finally, the authors showed that STIM2.3 expression significantly expands dendritic spine size compared with STIM2 expression. They postulated that regulated splicing of STIM2.3 may increase STIM2-mediated effects in neurons.

Overall, this manuscript presents many interesting findings comparing STIM2.3 with wildtype STIM2. Most of my concerns are about the interpretations of the results

1. Is STIM2.3 a gain-of-function variant as claimed by the authors? The findings with STIM2.3 may be better explained by a lack of negative regulation instead of "gain-of-function". It is unclear how STIM2.3 "gains" new functions by losing multiple regulatory motifs/sites. Also, STIM2.3 is not more potent than wildtype STIM2 in triggering NFAT translocation, a process depending on cytosolic calcium concentration.

2. Is STIM2.3 only present in old world monkeys and humans with expression in the brain? This conclusion was drawn by surveying a limited set of data. Based on the public database of GTExPortal (<https://www.gtexportal.org/>), RNA of STIM2.3 is present at a level similar to wildtype STIM2 in several human tissues outside of the brain, such as the spleen.

3. Based on both UniProt and NCBI databases, wildtype STIM2 contains 746aa instead of 833aa. STIM2.3 contains 599aa instead of 686aa.

4. The data supporting reduced association of AMPK with STIM2.3 (Figure 8A and 8B) are not convincing as there was less AMPK co-expressing with STIM2.3 in the input lane.

5. Further characterization of STIM2-d711 and STIM2-d675-710 may help understand negative regulation mechanisms of STIM2 within the C-terminal region upstream of SXIP motifs. There are many phosphoserine sites which may regulate STIM2 activation.

6. The titles are missing in all figures.

7. Figure2B: What does "24" and "30" mean?

8. The title "...is an evolutionary late gain-of-function regulator expressed in the brain" is confusing.

First revisionAuthor response to reviewers' comments**Reviewer 1: SUMMARY OF THE ADVANCE MADE IN THIS PAPER AND ITS POTENTIAL SIGNIFICANCE TO THE FIELD**

The manuscript "Alternatively-spliced STIM2.3 is an evolutionarily late gain-of-function regulator expressed in brain" by Poth et al details the discovery and molecular characterization of a novel alternatively-spliced isoform of the store-operated calcium entry (SOCE) channel activator STIM2. The authors demonstrate that the newly discovered splice variant, STIM2.3, is expressed in brain of late-evolving primates, and further show that STIM2.3 lacks 159 C-terminal amino acids but contains 12 unique amino acids compared to wildtype STIM2 (STIM2.2). Notably, the truncation of STIM2.3 includes the two EB1-binding domains and the poly-basic domain of wildtype STIM2.

Detailed structure-function analysis demonstrates that STIM2.3 is a gain-of-function variant of STIM2, though the exact mechanism of gain-of-function could not be delineated. The authors then show that STIM2.3 may play a role in increasing the size of neuronal dendritic spines with potential implications for synaptic strength and efficiency. These findings are significant because they add to our understanding of the molecular and functional complexity of the ubiquitous SOCE signaling mechanism, demonstrating that SOCE may be fine-tuned through alternative splicing in a cell and tissue-specific manner. These findings also shed new light on the role of SOCE in neurons and nervous system function overall, an area that has long been enigmatic in the calcium signaling field.

SUGGESTIONS TO AUTHORS

Overall, the experiments in this study are well-designed and executed, and the results support the conclusions as described. Most of my suggestions relate to enhancing the clarity of the writing and figures/data presentation and should not require new experiments (with the possible exception of Comment 1 below). The following are my specific suggestions and comments to the authors:

We thank both reviewers for their careful evaluation and insightful comments.

1. The legend for Figure 8 is missing, and the legend labeled "Figure 8" actually pertains to Figure 9. The lack of a legend also made it difficult to review and interpret the data presented in Figure 8. For example, in the pull-down experiments presented in 8A and B, it appears that pEYPC1 is used as an empty-vector control; however, the subsequent IPs utilize HA-tagged proteins and anti-HA as the pull-down antibody. A vector expressing HA alone would be a better empty-vector control instead of EYFP. Also, panel C for -thapsigargin quantifies total AMPK α whereas panel E for +thapsigargin (should this actually be panel D?) quantifies p-AMPK α . If there is a reason for quantifying total AMPK α versus p-AMPK α , it should be better explained. Lastly, the data in panels C and E would be easier to compare and interpret if the y-axes for both graphs were scaled the same.

We are sorry for the confusion we created. The Figures were indeed not correctly labeled. Due to space limitations, we now combined old Figures 7 and 8. We corrected the legends and added titles to all figure legends. Figure 8C (old) quantified the fraction of the total AMPK bound to the HA-STIM2 bead bound constructs. Originally, to normalize against potentially different amounts of bound HA-tagged constructs, we divided against total input (low HA input signal in A and B, multiplied by 40 to account for total protein input (usually 1mg), giving us low ratios on the Y-axis of the old figure. But we agree that it is indeed more relevant to normalize only the bound AMPK signal to the HA-bound bead fraction and we reanalyzed all existing and new experiments using volumetric analysis of either bound AMPK or bound phospho-AMPK over HA-bound signal, now giving us higher values and Y-axes that are comparable between bound AMPK and bound phospho-AMPK (with no change in findings, new Fig. 7 G,H). As the Thr172-phospho-specific AMPK antibody recognizes only the active form of AMPK, which can be activated by Ca²⁺ dependent CAMKK2, also reported to be in a trimeric complex with STIM2.2 (1), we also analyzed the amount of active AMPK bound. Again, the amount of bound activated AMPK is reduced with STIM2.3, which could indicate that due to the reduced interaction with AMPK, an increased Ca²⁺ influx through STIM2.3 is unable to compensate for the reduced binding (see also revised discussion). Interestingly, the amount of precipitated AMPK was not significantly affected by prior 10 min-long Thapsigargin treatment (grey points in Fig. 7 G,H), see also revised discussion. Additional blots are shown in Fig. S6 (blot transparency file).

Regarding the control with an HA-tagged protein, the reviewer is correct in that this is an important control. We had initially omitted this as Chauhan et al. already showed that deletion of STIM2 after the CAD domain obliterated interaction, however as they used a streptavidin binding tag, we repeated the experiments using both HA-STIM2.2, HA-STIM1 and HA-STIM1 with deletion before the CAD domain (STIM1 Δ 344). As seen in the Figure below, neither full length HA-STIM1 nor the HA-STIM1 deletion construct precipitated AMPK, while HA-STIM2.2 is very efficient at pulling down AMPK, confirming the specificity of the interaction and also confirming the inability of STIM1 to bind AMPK as reported by Chauhan et al (1). This information is now included in the text and in Suppl. Figure 4 (Fig. S4B).

Fig. S4B:

Control Blot showing Immunoprecipitation of HA-STIM1 Δ 344, HA-STIM1 or HA-STIM2.2 transfected constructs with endogenous AMPK α in HEK STIM1/2 $^{-/-}$ cells using anti-HA agarose. Membrane was incubated with the indicated antibodies and developed sequentially.

For the analysis shown in Figure 7 G, H (old Fig. 8 C,D) , we also redid statistical analysis by pairing resulting signals from individual blots which always had both HA-STIM2.2 and HA-STIM2.3 loaded, as also now described in the figure legend.

Overall, the results remain similar, STIM2.2 is more efficient at precipitating AMPK and both the results and discussion are amended with more detailed background and explanation.

As mentioned by reviewer #2, we also checked that expression of the STIM2 constructs did not affect the amount of endogenous AMPK expression, by normalization of input AMPK to β -actin, which we had also probed for on most blots (see also Fig. S4BC).

Fig. S4C: Quantification of AMPK input controls. (AMPK/ β -Actin)

2. In the graphs in Figures 3B,D and 4C,E, it is not immediately clear how the colors of the bars correspond to specific experimental conditions - it took me some time to figure out that the colors match those in the calcium traces in the preceding panels. I would suggest showing the same color key near the bargraphs as in the calcium traces, or label the bargraphs on the x-axes. In addition, it was not immediately clear that the title labels correspond to the graphs above (e.g., "resting" and "influx rate" in 3B). I would suggest moving the title to the tops of each graph.

We agree that the figure labels were not intuitive. We have also reformatted all bar graph analysis to better represent the spread of the data and replotted all analyzed data as box and whisker plots with the whiskers representing the min and max values including all data points (also included in figure labels and see methods). The traces in Figure 3 A,B now include numbers indicating which parameters were analyzed with the corresponding numbers in shown in 3 B,D. The x-axes in 3 B,D

and in 4C,E and following figures now also include labels and titles are moved to the top of the subpanels.

3. The microscopy images shown in Figures 5 and 6 would be easier to view and interpret if each single-channel image were shown in grayscale.

We changed the single-channel images to grayscale

4. Similar to above, nuclear vs cytoplasmic NFAT localization is difficult to determine in Figure 7A as shown. I would suggest separating the NFAT and STIM2 channels into component images shown in grayscale. And I would again suggest separating the images in Figure 9 into grayscale, single-channel images.

For Figure 7 changing the images in greyscales is possible, but it would result in adding 8 more subpanels to the figure as each channel has to be shown separately. Since STIM2.x is tagged with mKate and NFAT with GFP, we believe the colored overlay very clearly demonstrates that in the case of STIM2.2 at rest, a very efficient translocation of NFAT-GFP into the nucleus already took place, which is not the case for the other constructs, where translocation is seen upon Tg treatment. The images also show that no to very little translocation takes place for STIM2 Δ 5K and the images are thus representative of the statistical analysis shown in Fig. 7B.

5. Do the quantifications in Figure 6B and C represent the post-thapsigargin condition? This should be clearly indicated. Also, it is difficult to draw many conclusions from Figure 6C as shown other than that there are fewer 0.05 μ m² clusters for STIM2.2 compared to the other variants. It seems that there may also be differences for the large cluster sizes (0.6 - 1 μ m²) but the y-axis scale makes it impossible to clearly see this.

Yes, Fig. 6B and C are post Tg and we have now indicated this more clearly and also changed the single images into greyscales. To more clearly visualize the differences at the larger cluster sizes, we split Figure 7C into two domains with different Y-scales.

6. Several major concepts discussed throughout the paper are never thoroughly introduced, such that readers unfamiliar with the SOCE field may not understand their significance. Specifically, nuclear NFAT localization is analyzed as a readout of downstream SOCE function, but the role of SOCE in driving NFAT nuclear translocation is never explained. I would suggest including the role of SOCE in NFAT regulation in the Introduction. Also, many of the experiments test the role of the SxIP (EB1-binding) and poly-basic (PB) domains of STIM2, but the function of these domains is never fully explained.

We added more information and references within the introduction (line 51- 66) and in the relevant sections of the results (line 216, 223 and following), see marked copy of the text.

7. In line 75 (Results), the authors should clearly define that STIM2.2 corresponds to wildtype STIM2, and utilize the STIM2.2 notation throughout the paper.

We have changed the notation according to the reviewer's suggestion. (new line 113+ 123)

8. The sentence beginning on line 275, "Importantly, and unlike usually quantified,..." is unclear and should be reworded for clarity.

Paragraph has been rephrased for better clarity

9. It would be helpful if the authors would better describe autaptic cultures (i.e., how are they generated, what kind of cells do they include?).

additional text and a reference were added (line 301 -309)

10. Can the authors show representative images in Figure 9 of spines for control, STIM2.2, and STIM2.3 samples?

Are now included within old Figure 9, which is now renamed as Figure 8.

Reviewer 2: In this manuscript by Poth et al., the authors characterized an alternatively spliced variant of STIM2, named STIM2.3. In this STIM2.3 variant, the last 159 amino acids of wildtype STIM2 (STIM2.2) was replaced by 12 amino acids derived from Exon 13. As a result, several regulatory regions in STIM2, including the C-terminal polybasic motif, two microtubule end-binding SXIP motifs, and more than 10 potential serine phosphorylation sites, are missing in STIM2.3. The authors proposed that STIM2.3 is a "gain-of-function" variant based on a stronger thapsigargin-induced calcium response in STIM2.3-expressing cells compared with that in wildtype STIM2-expressing cells. Nevertheless, unlike wild-type STIM2, STIM2.3 was not able to promote NFAT nuclear localization in resting cells, a process dependent on STIM-mediated calcium entry. The authors further generated several STIM2 deletion mutants to test the role of missing regions in STIM2.3. Finally, the authors showed that STIM2.3 expression significantly expands dendritic spine size compared with STIM2 expression. They postulated that regulated splicing of STIM2.3 may increase STIM2-mediated effects in neurons.

Overall, this manuscript presents many interesting findings comparing STIM2.3 with wildtype STIM2. Most of my concerns are about the interpretations of the results

We thank both reviewers for their evaluation and insightful comments.

1. Is STIM2.3 a gain-of-function variant as claimed by the authors? The findings with STIM2.3 may be better explained by a lack of negative regulation instead of "gain-of-function". It is unclear how STIM2.3 "gains" new functions by losing multiple regulatory motifs/sites. Also, STIM2.3 is not more potent than wildtype STIM2 in triggering NFAT translocation, a process depending on cytosolic calcium concentration.

We agree that the term "GOF" can be discussed. In the widest sense it denotes an enhanced biological function of the gene product (which STIM2.3 shows in SOCE). In the case of NFAT, STIM2.3 initially indeed acts as a break compared with the basal high NFAT translocation seen in STIM1; STIM2 double deficient cells upon re-expression of STIM2.2. The finding that store-depletion-induced NFAT1 reached the maximal translocation (see Fig. 7A) for both STIM2 and STIM2.3 is due its relatively low overall Ca^{2+} threshold to translocate with full store-depletion (2, 3), obscuring a potential GOF phenotype, which, however, is seen for the overall amount of Ca^{2+} entry (AUC) during the time of NFAT translocation (Fig. 7 C,D, note that despite differences in AUC, NFAT translocation for STIM2.2, STIM2.3 and 2xIP- Δ 5K is similar). However, in the case of expression in neurons containing wild-type STIM2, we find that STIM2.3, but not STIM2.2 expression consistently increased the dendritic spine area, pointing towards either a specific or a GOF role (likely by specifically altering gene expression, potentially by differentially affecting NFAT4 or CREB). This function might also be achieved by an artificially deleted STIM2, thus indeed may be due to loss of a novel inhibitory region within the C-terminus that appears to be located between aa 588 and aa 623 (Δ 624 shows SOCE similar to wt), however, deletion of only this region from the full length STIM2.2 was unable to achieve increased function but rather slightly decreased STIM2 SOCE (Fig. S3).

To address the reviewer's point, we deleted the term "Gain of function" from the title and from parts of the text and added a more critical and detailed discussion

2. *Is STIM2.3 only present in old world monkeys and humans with expression in the brain? This conclusion was drawn by surveying a limited set of data. Based on the public database of GTExPortal (<https://www.gtexportal.org/>), RNA of STIM2.3 is present at a level similar to wildtype STIM2 in several human tissues outside of the brain, such as the spleen.*

The reviewer is correct in that STIM2.3 might not show exclusive expression in the brain, which, although we did not claim exclusive expression, is implied in our text. This is now written more carefully to include expression in other tissues (line 140 forward). We analyzed the GTExPortal carefully (<https://www.gtexportal.org/home/gene/STIM2#gene-transcript-browser-block>) and checking for isoform expression (see Fig. R1A below), detected ENST00000467011.6 (= STIM2.3) with TPM values above 2.2 in Cerebellum, brain Cerebellar Hemisphere, Cortex, but also in

Prostate, Pituitary and Spleen on the gTExPortal under the tab ‘isoform expression’ (Fig. R1A, arrows).

However, when using the tab ‘Junctions’ to detect reads into and out of the alternative exon, we only detect junction 20 (splicing to alternative exon) and junction 23 (out of exon 2.3) in brain (see Fig. R2, B), with the only exception being skeletal muscle (which shows no TPM reads in the isoform tab) and no junctional reads for Spleen, Prostate or Pituitary (Fig R1B), uncovering an apparent inconsistency within the conventional RNA-seq data (from 2011) used to generate the plots.

Fig. R1 A, B: Data derived for STIM2 isoform and junctional reads from the GTExPortal (weblink see as in Figure). Arrows refer to the tissues mentioned in our reply above.

To gain more insight into this, we decided go back to original raw RNAseq data and process it ourselves using several splice detection tools. However, it turned out that getting access to the raw data behind GTEx involves a complicated application process. Instead, we used openly accessible data from the ENCODE portal that provides raw RNAseq data for a good amount of human tissues.

In the light of the GTEx results for STIM2.3, we obtained RNA-Seq data for skeletal muscle (arm, leg and back) and spleen from ENCODE. We then performed isoform-level (used tools: Salmon, Kallisto) and junction-level (used tool: rMATS) quantification, see Fig. R2 below, parts of which are not shown as new Figure S1. This analysis showed the following: i) ENST00000467011.6 (STIM2.3) is expressed in skeletal muscle and spleen ii) Exon 13 (STIM2.3-specific exon) is differentially included in skeletal muscle. We observe the same discrepancy between isoform and junction-level information on the GTEx portal. An explanation for this may be that isoform-level quantification methods such as Salmon and Kallisto do not distinguish clearly between reads belonging to longer and shorter versions of the exon 13 found in ENST00000477474.3 and STIM2.3, respectively.

We included parts of Figure R1 as new Figure S1 and additional sentences (results, line 140 forward, discussion line 380 forward) describing this discrepancy and were more careful in our wording.

Fig.R1 A, B, C: Data derived for STIM2 isoform and junctional reads using data from ENCODE.

STIM2.3 and exon 13 of STIM2.3 are highlighted. (A) and (B) reflect isoform quantification by the tools kallisto and salmon. (C) reflects junction-level quantification by the tool rRMATS.

Further evidence confirming the postulated evolutionarily late expression of the alternate exon of STIM2.3 in brain is also provided by the report of Recinos et al.: Lineage-specific splicing regulation of MAPT gene in the primate brain. *Cell Genom*, 2024 (4), which modeled cassette exon inclusion in primate brains as a quantitative trait and identified 1,170 (~3%) exons with lineage-specific splicing shifts under stabilizing selection, based on RNA-seq data from brains of human and six other primate species. Among the exons with lineage-specific splicing shifts, the alternate exon underlying STIM2.3 (27022623-27022690, human Refgenome hg19) confirmed 1. our average splice probability (20%) in human brain and 2. A significant upregulation in human vs. non-human and in hominoids vs. non-hominoids (see Suppl. Table 1 in Recinos et al).

In addition, we added analysis of different patient-derived classified glioblastoma (GB) samples of either neuronal-, mesenchymal- or classic GB origin (line 170 onward), where we detected significant STIM2.3 expression only in the neuronal derived GB samples.

3. Based on both UniProt and NCBI databases, wildtype STIM2 contains 746aa instead of 833aa. STIM2.3 contains 599aa instead of 686aa.

ENST00000698882.1 translates to Uniprot A0A8V8TMC8_HUMAN, which corresponds to STIM2.2 with 833 amino acids. The difference is due to the unconventional long signal peptide (SP, starting with MNA see also NCBI NM_020860.4 with "misc_feature 118..120

```

/gene="STIM2"
/note="upstream_AUG_codon; putative N-terminal extension:
MNAAGIRAPEAAGADGTRLAPGGSPCLRRRGRPEESPAAVVAPRGAGELQAAGAPLRF
HPASPRRLHPASTPGPAWGWLLRRRRWAA" (87 amino acids)

```

This putative N-terminal extension explains the difference in amino acids and corresponds to a protein of 93 kDa.

Graham et al., *JBC* 2011 (5) found that fusing the long STIM2 SP (Met1-Gly101) to the constant Fc region of human IgG resulted in ER dependent glycosylation and secretion of the Fc fusion protein, but not an abbreviated Leu88-Gly101 STIM2 signal peptide, proving the functionality of the unusually long signal peptide and in part contradicting Williams et al (6). In contrast to the artificial short STIM2 SP, the short STIM1 SP is sufficient for ER targeting and indeed nearly all investigations into heterologous STIM2 function utilize constructs with the STIM1 SP.

We have compared expression and function of STIM2 with the short STIM1SP with that of STIM2 with its long endogenous signal peptide and found no functional difference regarding SOCE (Fig. R3).

Fig. R3: Comparison of Fura2 based Ca²⁺ imaging of N-terminal YFP tagged STIM2 containing the STIM1 signal peptide (S1SP-YFP-STIM2.2) with a C-terminal eGFP tagged STIM2.2 with its endogenous 101 aa signal peptide starting at "MNA" (named STIM2.2-eGFP within the figure).

However, as STIM2 Uniprot ID Q9P246-1 is indeed often defined as the conventional variant, in which the non-AUG codon (UUG) Leucine is chosen to represent the starting Methionine (NP_065911.3) (6), yielding a protein of 746 aa, we added both numbers to Figure 1 but changed all

subsequent numbering to correspond to the defined conventional variant, as also explained in the manuscript (line 113 and following).

4. The data supporting reduced association of AMPK with STIM2.3 (Figure 8A and 8B) are not convincing as there was less AMPK co-expressing with STIM2.3 in the input lane.

In all cases we investigated the amount of bound endogenous AMPK from the transfected HEKDKO (Stim deficient) cells with the input showing endogenous AMPK. In the sample blot shown in old Fig. 8A (now new Fig. 7F), it may appear that there is less AMPK in some of the input lanes. We therefore reanalyzed all blots in which we had also probed with a β -Actin antibody, but which was not shown in old Figure 8 (see blots in Fig. S6, blot transparency file and also new Fig. S4B, where we performed an additional control experiment using HA-tagged STIM1 constructs, see also comments of reviewer #1). New Fig. S4C shows this quantification with the individual pEYFP transfections set to 100%. Expression of the variants on average does not significantly change the amounts of endogenous AMPK. These data are now included in the results section (line 316-319). For more clarity, the results section also now contains more background information.

Added to Supplementary Figure 4:

B. Immunoprecipitation of HA-STIM1 Δ 344, HA-STIM1 or HA-STIM2.2 transfected cells with endogenous AMPK α in HEK STIM1/2 cells using anti-HA agarose. Membrane was incubated with the indicated antibodies and developed sequentially.

C. Analysis of AMPK-input signals normalized to β -Actin signal of transfected cells analyzed in Fig. 8 (new Fig. 7F).

5. Further characterization of STIM2-d711 and STIM2-d675-710 may help understand negative regulation mechanisms of STIM2 within the C-terminal region upstream of SXIP motifs. There are many phosphoserine sites which may regulate STIM2 activation.

Yes, we agree that further point mutations of the phosphorylation sites especially in the region between STIM2.2 aa 588 and 623 would be very interesting. As STIM2.2 deletion at aa 624 (Fig. S2) has the same SOCE as STIM2.2 and is smaller than the STIM2.3 mimicking deletion at aa 587 (Fig. 4), the simplest explanation would be the location of an inhibitory signal between aa 588 and 623. However, deletion of this region from full length STIM2.2 (see Fig. S3 B,C) did not increase, but slightly decreased SOCE compared to STIM2.2, possibly due to C-terminal misfolding. Within this region there are indeed 8 potential phosphorylation sites (S599, S603, S609, L612, S613, Y617, T619 and S621, as detected by www.phosphosite.org). Mutations of each one or all of these might uncover the true reason for the increased SOCE function of STIM2.3, but was not possible in this study and in the time for revision of the paper. However, we believe that these sites in the context of cancer (as most of the sites were detected in phospho-proteome screens of cancer tissues) would be interesting to pursue in the future.

6. The titles are missing in all figures.

We are very sorry for this omission. They are now added.

7. Figure2B: What does "24" and "30" mean?

These are the number of junctional reads derived from RNA-seq analysis. We specified this more clearly in the figure legends. We also included a discussion of discrepancies between junctional reads and exon usage derived from RNA-Seq experiments (see response to question #2).

8. The title "...is an evolutionary late gain-of-function regulator expressed in the brain" is confusing.

We changed the title to read: **Alternatively-spliced STIM2.3 is an evolutionary late Store-operated Calcium Entry regulator expressed in brain.**

References for "reply to reviewers":

1. A. S. Chauhan *et al.*, STIM2 interacts with AMPK and regulates calcium-induced AMPK activation. *FASEB J* **33**, 2957-2970 (2019).
2. P. Kar, C. Nelson, A. B. Parekh, Selective activation of the transcription factor NFAT1 by calcium microdomains near Ca²⁺ release-activated Ca²⁺ (CRAC) channels. *Journal of Biological Chemistry* **286**, 14795-14803 (2011).
3. P. Kar, A. B. Parekh, Distinct spatial Ca²⁺ signatures selectively activate different NFAT transcription factor isoforms. *Mol Cell* **58**, 232-243 (2015).
4. Y. Recinos *et al.*, Lineage-specific splicing regulation of MAPT gene in the primate brain. *Cell Genom* **4**, 100563 (2024).
5. S. J. Graham, M. A. Dziadek, L. S. Johnstone, A cytosolic STIM2 preprotein created by signal peptide inefficiency activates ORAI1 in a store-independent manner. *J Biol Chem* **286**, 16174-16185 (2011).
6. R. T. Williams *et al.*, Identification and characterization of the STIM (stromal interaction molecule) gene family: coding for a novel class of transmembrane proteins. *Biochem J* **357**, 673-685 (2001).

Second decision letter

MS ID#: jcs.264353R1

MS Title: Alternatively-spliced STIM2.3 is an evolutionary late Store-operated Calcium Entry regulator expressed in brain.

Authors: Vanessa Poth; Hoang Thu Trang Do; Lukas Jarzembowski; Katrin-Lisa Laius; Kathrin Foerderer; Thomas Tschernig; Hanah Betsy Robertson; Dalia Alansary; Reza Shaebani; Volkhard Helms; Barbara A Niemeyer

Article Type: Research Article

Dear Dr Niemeyer,

Thanks for submitting your revised article to JCS. I am happy to tell you that your manuscript has now been accepted for publication in Journal of Cell Science, pending standard publication integrity checks.